# MVSFormer: Multi-View Stereo by Learning Robust Image Features and Temperature-based Depth

**Chenjie Cao**                                                 *20110980001@fudan.edu.cn*
*School of Data Science, Fudan University*

**Xinlin Ren**                                                  *20110240015@fudan.edu.cn*
*School of Data Science, Fudan University*

**Yanwei Fu**[*]                                                *yanweifu@fudan.edu.cn*
*School of Data Science, Fudan University*

**Reviewed on OpenReview:** *https://openreview.net/forum?id=2VWR6JfwNo*

## Abstract

Feature representation learning is the key recipe for learning-based Multi-View Stereo (MVS). As the common feature extractor of learning-based MVS, vanilla Feature Pyramid Networks (FPNs) suffer from discouraged feature representations for reflection and texture-less areas, which limits the generalization of MVS. Even FPNs worked with pre-trained Convolutional Neural Networks (CNNs) fail to tackle these issues. On the other hand, Vision Transformers (ViTs) have achieved prominent success in many 2D vision tasks. Thus we ask whether ViTs can facilitate feature learning in MVS? In this paper, we propose a pre-trained ViT enhanced MVS network called MVSFormer, which can learn more reliable feature representations benefited by informative priors from ViT. The finetuned MVSFormer with hierarchical ViTs of efficient attention mechanisms can achieve prominent improvement based on FPNs. Besides, the alternative MVSFormer with frozen ViT weights is further proposed. This largely alleviates the training cost with competitive performance strengthened by the attention map from the self-distillation pre-training. MVSFormer can be generalized to various input resolutions with efficient multi-scale training strengthened by gradient accumulation. Moreover, we discuss the merits and drawbacks of classification and regression-based MVS methods, and further propose to unify them with a temperature-based strategy. MVSFormer achieves state-of-the-art performance on the DTU dataset. Particularly, MVSFormer ranks as Top-1 on both intermediate and advanced sets of the highly competitive Tanks-and-Temples leaderboard. Codes and models are released in `https://github.com/ewrfcas/MVSFormer`.

## 1  Instruction

Multi-View Stereo (MVS) aims to reconstruct highly detailed 3D representations with multi-view images, with the key step of estimating depth maps with known camera poses (Furukawa & Hernández, 2015). Many traditional methods (Barnes et al., 2009; Furukawa & Ponce, 2009; Galliani et al., 2015; Schönberger et al., 2016) successfully make use of matching low-level features of images; unfortunately they may be negatively affected by various occlusions and different illumination conditions. To this end, learning-based methods enhanced by Deep Neural Networks (DNNs) (Ji et al., 2017; Yao et al., 2018; Gu et al., 2020; Giang et al., 2021; Wei et al., 2021) have developed recently. Typically, there are three steps for learning-based MVS methods, *i.e.*, feature extraction, cost volume construction, and cost volume regularization (Yao et al., 2018).

---

[*]Corresponding author.

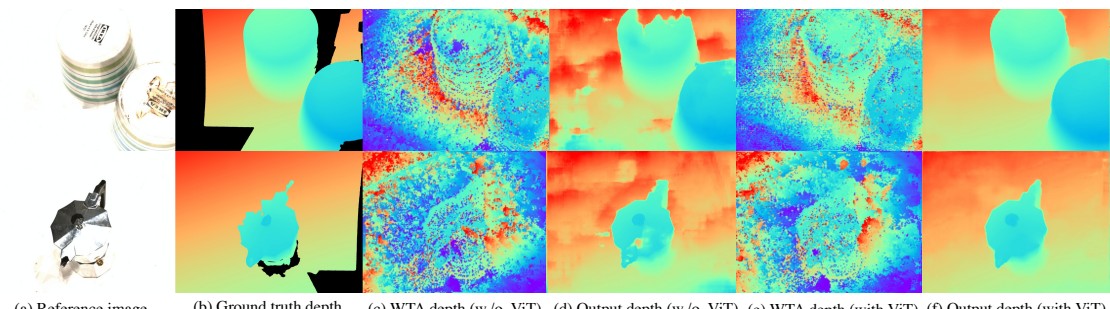

Figure 1: Hard cases in DTU (Aanæs et al., 2016) with reflection and texture-less regions. (c)(d) indicate Winner-Take-All (WTA) depth from the feature correlation (1/8 scale) achieved by the dot products among reference and source features (Eq. 4) before the 3D CNN cost volume regularization. WTA depth enhanced with pre-trained ViT (Twins-small (Chu et al., 2021a)) contains less noise compared with one without ViT priors. (f) can also get better final depth predictions compared with (d).

Many works devote to formulating better cost volumes with efficient multi-stage models (Gu et al., 2020; Yang et al., 2020; Mi et al., 2021) and visibility information (Zhang et al., 2020; Xu & Tao, 2020). Meanwhile, learning a superior cost volume regularization with hybrid 3D U-Net structures (Luo et al., 2019; Sormann et al., 2020) or Recurrent Neural Networks (RNNs) (Yao et al., 2019; Wei et al., 2021; 2022) is also shown to improve performance. Generally, the cost volume regularization is designed to *refine* noise-contaminated cost volumes with non-Lambertian surfaces or object occlusions (Yao et al., 2018) to smooth feature correlations as shown in Fig. 1(c)(e). Such regularization can not completely rectify ambiguous feature matchings from reflections or texture-less regions with unreliable 2D image features. Therefore, it is still of great significance to learn good representative features during the feature extraction to improve the generalization of MVS.

As a common solution for feature extraction, Feature Pyramid Network (FPN) (Lin et al., 2017a) learns multi-scale image features in most MVS networks. Some works leverage deformable convolutions (Wang et al., 2021a; Wei et al., 2021; Ding et al., 2021), attention mechanisms (Yi et al., 2020; Zhu et al., 2021; Ding et al., 2021), and normal curvatures (Xu et al., 2020; Giang et al., 2021) to learn more reliable features for FPNs. Nevertheless, these works still suffer from poor generalization in modeling the reflection and texture-less regions as visualized in Fig. 1(c)(d). On the other hand, few previous efforts have explicitly exploring the features from the Convolutional Neural Networks (CNNs) pre-trained on extra image data (Ding et al., 2022), *e.g.*, ResNet (He et al., 2016), as such pre-trained CNNs may have some problems in MVS: 1) *low-level features of CNNs only consider limited receptive fields*, which lack the holistic image understanding, and fail to tackle with reflections and texture-less areas; 2) *high-level features of CNNs are of highly semantic abstract*, thus best for the classification rather than the fine-grained visual feature matching. We empirically validate that pre-trained CNN models fail to achieve significant improvement in MVS as in Tab. 4.

Recent Vision Transformers (ViTs) have achieved impressive performance on various image understanding tasks (Dosovitskiy et al., 2020; Caron et al., 2021; Wang et al., 2021b; Liu et al., 2021; Chu et al., 2021a; He et al., 2021). Thus, a nature question is *whether we can significantly strengthen the feature representation learning of MVS with pre-trained transformers from external 2D image dataset?* For the issues of reflections and texture-less regions in MVS, ViTs equipped with long-range attention modules can provide global understanding for MVS models rather than the low-level textures. Moreover, the patch-wise feature encoding of ViTs works reasonably well for feature matching (Sun et al., 2021; Jiang et al., 2021). Since the depth prediction is intrinsically a 1D feature matching problem along epipolar lines (Furukawa & Hernández, 2015), ViTs shall be the recipe for learning-based MVS. Unfortunately, to the best of our knowledge, there is no work explicitly exploiting pre-trained ViTs for MVS.

To this end, we thoroughly explore how to make pre-trained ViTs boost the MVS performance. As shown in Fig. 1(e)(f), features from pre-trained ViT are complementary to those from FPN, and facilitate better modeling reflection and texture-less regions in MVS. Formally, we propose using the pre-trained ViTs to enhance FPN for feature extraction in MVS, and further formulate a novel MVS Transformer (MVSFormer). Specifically, we employ the hierarchical ViT Twins (Chu et al., 2021a) as the backbone of MVSFormer.

Benefited by the pyramid architecture and the efficient attention mechanism, MVSFormer can be trained in high-resolution and typically achieve better results compared with FPNs. Moreover, we also extend the alternative MVSFormer to plain-ViT with vanilla attention (Dosovitskiy et al., 2020; Caron et al., 2021; He et al., 2021), called MVSFormer-P. Compared with using efficient and hierarchical ViTs, we freeze the *off-the-shelf* ViT backbone pre-trained by the self-distillation method – DINO (Caron et al., 2021) in MVSFormer-P[1] during the training to reduce the computation as in Fig. 2(A). Note that the MVSFormer-P still enjoys superior performance compared with other state-of-the-art MVS methods.

Further, we present an efficient multi-scale strategy to train MVSFormer, as it is non-trivial to directly train ViTs for MVS. In MVS tasks, the models should be tested on various high-resolution images (Aanæs et al., 2016; Knapitsch et al., 2017), while they have to be trained on low-resolution to save the training computations. It is thus not amenable to directly extending pre-trained ViTs without scale invariance to MVS tasks. Our empirical experiments in Sec. 4.1 show that the proposed multi-scale strategy is sufficient to generalize MVSFormer to test images with much higher resolutions (1536 or 1920) compared with the largest one during the training (1280).

Furthermore, we technically unify the advantages of both regressive depth (regression), and *argmax* depth (classification) for MVSFormer. Different from previous works (Wang et al., 2022a; Peng et al., 2022; Wang et al., 2022b), we find that optimizing the MVS network with cross-entropy loss can achieve much more reliable confidence maps but slightly worse depth predictions. Because the *argmax* operation can not provide exact depth results, which harms the depth performance. To address this issue, MVSFormer predicts the temperature-based depth expectation instead of the *argmax* during the inference and achieves smooth depth predictions and superior final point clouds.

Our contributions can be highlighted as: (1) To the best of our knowledge, it is the first work that systematically explores the influence of pre-trained ViTs on MVS. Learning a better feature representation by the feature extractor is important to set up a bridge between 2D vision and 3D MVS tasks. (2) We propose a novel ViT enhanced MVS network – MVSFormer, which is further trained with the efficient multi-scale training strategy to be generalized for various resolutions. The proposed Twins-small pre-trained MVSFormer remarkably reduces the overall error of point cloud reconstruction in DTU from 0.312 to 0.289 compared with the CNN-based pre-trained ResNet with competitive computations and all other model settings unchanged as in Tab. 4. (3) We analyze the merits and limitations of regression and classification-based MVS, and propose a simple but effective way to unify both.Empirical qualitative evidences in Fig. 4 and Fig. 6 show that classification-based confidence can filter outliers for the real-world reconstruction. And quantitative results in Tab. 6 indicate that our temperature-based depth predictions also enjoy superior point cloud metrics. (4) The proposed methods can achieve the state-of-the-art performance in DTU dataset (Aanæs et al., 2016), Tanks-and-Temples (Knapitsch et al., 2017), and ETH3D (Schops et al., 2017).

## 2 Related Works

**Learning-based MVS methods.** Learning-based MVS methods enhanced with DNNs have been extensively studied for MVS tasks (Ji et al., 2017) . MVSNet (Yao et al., 2018) proposes an end-to-end pipeline based on a 2D CNN image feature extraction, cost volumes formulated by homography warping, and a cost volume regularization by 3D CNN. Many works are devoted to reducing the heavy computation of 3D CNN-based cost volume regularization with coarse-to-fine strategies (Gu et al., 2020; Yang et al., 2020; Mi et al., 2021) and RNNs (Yao et al., 2019; Wei et al., 2021; 2022). Meanwhile, some researches formulate a more reliable cost volume, such as the visibility of ViS-MVSNet (Zhang et al., 2020), and the epipolar aggregation of MVSTER (Wang et al., 2022b). Besides, many works try to learn a better cost regularization by hybrid 3D U-Net (Luo et al., 2019; Sormann et al., 2020), RNN-3DCNN (Wei et al., 2021), and epipolar attention (Ma et al., 2021; Yang et al., 2021). Although many efforts are paid to achieve better cost volumes, we advocate that learning superior feature representations are more effective for a generalized MVS method. Our method

---

[1]Intuitively Twins can also be pre-trained by the self-supervised task as DINO. However, it is non-trivial to implement and train it, due to technical difficulty and expensive running costs on ImageNet (taking 1 month on 2 V100 GPUs). Thus the self-supervised Twins is beyond the scope of this paper.

explores using pre-trained ViTs to improve MVS feature learning, which is orthogonal to these works based on cost volumes.

**Feature learning in MVS.** Since 2D image features are critical in MVS learning, powerful FPN (Lin et al., 2017a) is the common solution. FPN is designed as a U-Net to fuse multi-scale features. Deformation convolutions (Dai et al., 2017) are widely used in MVS to improve the receptive fields flexibly (Wang et al., 2021a; Wei et al., 2021; Mi et al., 2021; Ding et al., 2021). Furthermore, Xu et al. (2020) and Giang et al. (2021) leverage fixed and learnable normal curvatures to dynamically select kernel sizes for FPN, which can learn robust features for various image resolutions. Besides, the attention mechanism is also utilized for MVS feature learning. Many works leverage intra- or inter-attention during the feature learning (Yi et al., 2020; Zhu et al., 2021; Ding et al., 2021) for long-range feature dependencies. As these efforts exactly improve the performance of the FPN, we should notice that such features still have inductive biases from CNNs, thus leading to inferior performance to patch-wise ViTs (Dosovitskiy et al., 2020). There is no systematical exploration of pre-trained ViT models to MVS feature learning.

**Vision Transformers.** Self-supervise pre-trained transformers have achieved prominent success in Natural Language Processing (NLP) (Vaswani et al., 2017; Devlin et al., 2018; Brown et al., 2020). Inspired by these achievements, transformers are also introduced into the computer vision (Dosovitskiy et al., 2020; Liu et al., 2021; He et al., 2021). ViTs achieve state-of-the-art performance in many vision tasks, which include image classification (Dosovitskiy et al., 2020; Liu et al., 2021), detection and segmentation (Zheng et al., 2021; Li et al., 2022), and so on. Moreover, task specific transformers also achieve successes in optical flow (Huang et al., 2022) and point cloud processing (Guo et al., 2021; Zhao et al., 2021). Compared with CNNs, ViTs enjoy much more long-range modelling capacity. Since ViTs lack some inductive biases inherent to CNNs, such as translation equivalence and locality, they are much more data-hungry to generalize to unseen data (d'Ascoli et al., 2021; Dosovitskiy et al., 2020; Xu et al., 2021). Therefore, pre-training is necessary for ViTs. For different pre-training tasks, ViTs can be categorized into self-supervised ones (Bao et al., 2021; He et al., 2021; Caron et al., 2021) and supervised ones (Dosovitskiy et al., 2020; Liu et al., 2021; Chu et al., 2021a). Supervised ViTs are usually pre-trained for the classification task, while self-supervised ones are implemented with masked prediction (He et al., 2021; Bao et al., 2021) or contrastive learning (Caron et al., 2021; Chen et al., 2021b). Furthermore, to reduce notorious computations and memory costs of the vanilla attention, many ViTs use multi-scale architectures (Liu et al., 2021; Wang et al., 2021b; Chu et al., 2021a) instead of single-scale backbones. Particularly, many works tried to finetune pre-trained ViTs with task specific components, such as FPNs or CNNs, for various downstream tasks (Li et al., 2022; Huang et al., 2022; Chen et al., 2021a). Although pre-trained ViTs are appealing in many fields, releasing their potential in MVS is non-trivial, which will be discussed in the following sections.

**Multi-scale training.** It is well known that CNNs can be generalized to higher resolutions for high-level vision tasks such as classification (Chu et al., 2021b), which is benefited from their scale invariance. Some classical object detection methods (Cai et al., 2016; Redmon & Farhadi, 2017) emphasize the importance to use multi-scale training for better robustness to various image scales. Further, some recent generation works (Dong et al., 2022; Chai et al., 2022; Cao et al., 2022) show that multi-scale training is also a data and computation efficient way to improve the generation quality at high resolution. For the MVS learning, dynamic kernels enhanced by the curvature clues (Giang et al., 2021) are leveraged to tackle the image scale gaps for CNN-based models. Unfortunately, the attention-based ViTs generally do not have the translation equivalence and locality; and these make ViTs more easily overfitting to specific resolution or sequence length than CNNs (d'Ascoli et al., 2021; Dosovitskiy et al., 2020; Xu et al., 2021). Although Conditional Positional Encoding (CPE) and zero-padding are utilized to alleviate the spatial overfitting of ViT (Chu et al., 2021b), multi-scale training is still adopted in many ViTs for high-resolution detection tasks (Liu et al., 2021; Chu et al., 2021a; Carion et al., 2020). In this paper, we empirically evaluated that multi-scale training is the key strategy to release the potential of ViTs for MVS in Sec. 4.2. Technically, we repurpose the gradient accumulation to improve GPU utilization for better efficiency of the multi-scale training.

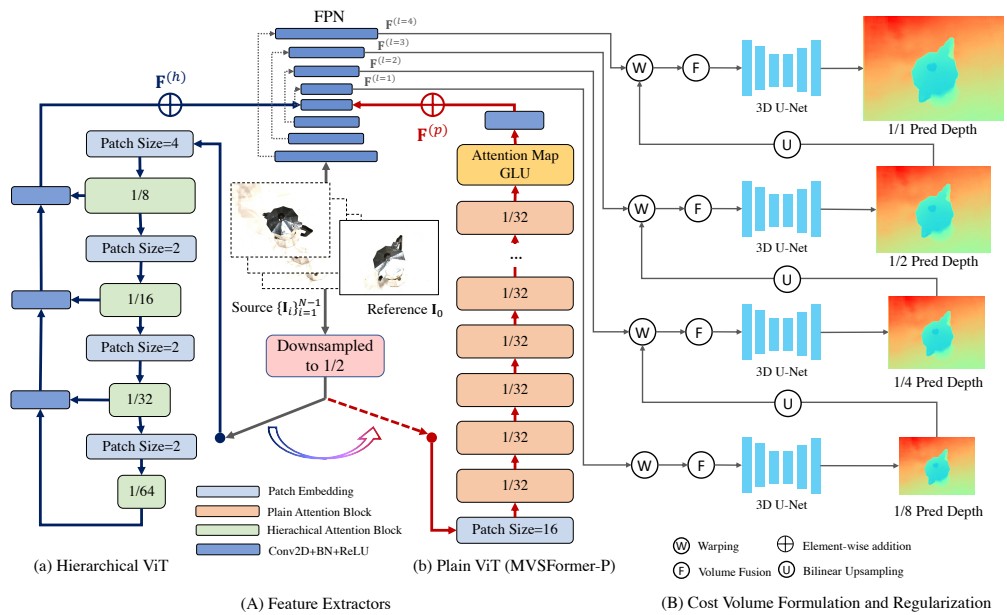

Figure 2: The overview of MVSFormer. (A) Feature extractors of hierarchical ViT (a) and plain ViT (b). Inputs for ViTs are downsampled to the 1/2 resolution. (B) Multi-scale cost volume formulation and regularization. 'Warping': warping source features with upsampled depth hypotheses (Eq. 3) for cost volumes (Eq. 4). 'Volume Fusion': fusing cost volumes from all source views with respective visibility (Eq. 5).

## 3 Method

**Overview.** The architecture of MVSFormer is overviewed in Fig. 2, as the seminal MVS pipeline in Yao et al. (2018). Given a group of $N$ images with different views which contain reference image $\mathbf{I}_0 \in \mathbb{R}^{H \times W \times 3}$ and source images $\{\mathbf{I}_i\}_{i=1}^{N-1} \in \mathbb{R}^{H \times W \times 3}$, as well as the their camera intrinsics and extrinsics, MVSFormer learns feature representation in feature extraction (Sec. 3.1), enhanced by the hierarchical ViT– Twins (Chu et al., 2021a) (Fig. 2(a)) or the plain ViT– DINO (Caron et al., 2021) (Fig. 2(b)) with several novel training strategies (Sec. 3.2). Then the multi-stage cost volume formulation and regularization are presented to compute the probabilities of coarse-to-fine depth hypotheses (Sec. 3.3). Finally, cross-entropy loss is employed to optimize the MVSFormer, while the inference is made on depth expectation (Sec. 3.4).

**Preliminaries**. (1) *Twins* (Chu et al., 2021a) is pre-trained in a supervised way by the hierarchical-ViT model as in Fig. 2(a). To further reduce the complexity, Twins employs the separable locally-grouped self-attention and global sub-sampled attention to construct each attention block. Such a global and local design outperforms the classic pyramid ViT (Wang et al., 2021b). Twins also leverages CPE (Chu et al., 2021b) with 2D depthwise convolutions instead of the absolute positional encoding in other ViTs (Dosovitskiy et al., 2020). (2) *DINO* (Caron et al., 2021) is pre-trained in a self-supervised way by the self-distillation of plain-ViT as in Fig. 2(b). The prominent characteristic of DINO is that attention maps of its last layer can learn class-specific features leading to unsupervised object segmentations as discussed in Caron et al. (2021), and shown in Fig. 3. Thanks to the unsupervised training and the multi-crop strategy, the feature representations of DINO can be well generalized to various environments, illuminations, and resolutions.

### 3.1 Feature Extraction

As many MVS works, we also use FPN (Lin et al., 2017a) as the main feature extractor, which is enhanced with pre-trained ViTs. In MVSFormer, ViTs work to formulate global feature correlations, while FPN is devoted to learning detailed ones. Before taking reference and source images to the ViT, we first downsample them into $(\frac{H}{2}, \frac{W}{2})$ to save the computation and memory costs. We resize the absolute position encoding of pre-trained ViTs with bicubic interpolation to fit different image scales (Dosovitskiy et al., 2020). Then hierarchical-ViT output $\mathbf{F}^{(h)}$ or plain-ViT output $\mathbf{F}^{(p)}$ is directly added to the highest level feature of the

FPN encoder. Thus we can get coarse-to-fine features $\{\mathbf{F}^{(l)}\}_{l=1}^{L=4}$ from the FPN decoder scaled from $(\frac{H}{8}, \frac{W}{8})$ to $(H, W)$ of the origin resolution as shown in Fig. 2(A). These features contain priors from both ViT and CNN, and will be leveraged to formulate more reliable cost volumes. We have tried other feature fusion strategies in Sec. C.2 of the Appendix, but the difference is negligible. So the simple but effective feature addition is adopted in MVSFormer.

**MVSFormer with trainable Twins.** The Twins is used as the backbone in our default MVSFormer without additional explanations, because it enjoys the best reconstruction performance as in Sec. 4.1. To finetune MVSFormer in various resolutions, the ViT backbone should meet two conditions, *i.e.*, the efficient attention mechanism and the robust position encoding for different scales, which are both solved by Twins elegantly. Except for the pyramid architecture, CPE in Twins can learn positional cues from the zero-padding (Islam et al., 2020), and breaks the permutation-equivalent of ViTs with proper CNN inductive biases. As shown in Fig. 2(b), MVSFormer encodes 4 multi-scale features $\{\mathbf{F}^{(h,s)}\}_{s=1}^{S=4}$ with $(\frac{1}{8}, \frac{1}{16}, \frac{1}{32}, \frac{1}{64})$ of the origin resolution respectively. We use another FPN to upsample these multi-scale features as

$$\mathbf{F}^{(h)} = \text{FPN}(\mathbf{F}^{(h,1)}, \mathbf{F}^{(h,2)}, \mathbf{F}^{(h,3)}, \mathbf{F}^{(h,4)}) \in \mathbb{R}^{\frac{H}{8} \times \frac{W}{8} \times C}. \tag{1}$$

Benefited by the efficient attention design, we can finetune the pre-trained Twins during the training phase with a relatively low learning rate in various resolutions. More details are discussed in Sec. 4.

**MVSFormer-P with frozen DINO.** We also explore learning MVSFormer based on plain-ViT with vanilla attention, *i.e.*, MVSFormer-P. Although plain ViTs suffer from heavy computations for the high-resolution MVS learning, we find that the self-supervised ViT – DINO enjoys good properties to improve the feature representation with even frozen ViT backbones. Particularly, attention maps of $[CLS]$ token from the last layer of DINO as in Fig. 3 are utilized to strengthen the feature learning. Since MVS is a feature matching task essentially, the priors of object or scene segmentations are useful to avoid confused depth predictions of foreground and background. As inputs to ViT have been halved yet, after the patch-wise embedding with kernel and stride size 16, DINO performs the attention-based learning for feature maps with the resolution of $(\frac{H}{32}, \frac{W}{32})$. To better utilize the good segmentation property of DINO, we use a trainable Gated Linear Unit (GLU) (Shazeer, 2020) to reduce the dimension of DINO features $\mathbf{F}_{dino}$. We assume that $\mathbf{A} \in \mathbb{R}^{\frac{H}{32} \times \frac{W}{32} \times h}$ indicates attention maps of the $[CLS]$ token with $h$ attention heads from the last layer of DINO. $\hat{\mathbf{A}} \in \mathbb{R}^{\frac{H}{32} \times \frac{W}{32} \times 1}$ is averaged along $h$ heads from $\mathbf{A}$. Thus GLU can be written as

$$\tilde{\mathbf{F}}^{(p)} = \text{Swish}(\text{ConvBN}_l([\mathbf{F}_{dino}; \mathbf{A}])) \odot \text{Swish}(\text{ConvBN}_r([\mathbf{F}_{dino} \odot \hat{\mathbf{A}}])) \in \mathbb{R}^{\frac{H}{32} \times \frac{W}{32} \times C_p}, \tag{2}$$

where $\text{Swish}(x) = x \cdot \text{sigmoid}(x)$ (Ramachandran et al., 2017); $\odot$ means the element-wise multiplication; $[\cdot; \cdot]$ indicates the concat operation; $C_p$ is the reduced channel of DINO. GLU helps to protect important features benefited from the segmentation attention maps during the dimension reduction, which can effectively improve the MVS performance. Then we use two transpose convolutions to upsample $\tilde{\mathbf{F}}^{(p)}$ to $\mathbf{F}^{(p)} \in \mathbb{R}^{\frac{H}{8} \times \frac{W}{8} \times C}$ for the feature addition to the FPN encoder with channels $C$. Although the plain-ViT based DINO suffers from heavy memory and computation costs, we find that MVSFormer-P can still work well with trainable GLU and upsample convolutions when DINO is frozen. Therefore, the alternative MVSFormer-P only demands a little more memory cost compared with the vanilla MVS FPN training; and achieves competitive results as the full trainable MVSFormer.

## 3.2 Efficient Multi-scale Training

Although ViTs has large capacity, missing translation equivalence and locality make them vulnerable to handling various input resolutions (Xu et al., 2021). Unfortunately, most MVS tasks should be tested in different High-Resolution (HR) (from 1200×1600 (Aanæs et al., 2016) to 1080×1920 (Knapitsch et al., 2017)). CNN-based methods can largely solve this problem with dynamic kernels (Giang et al., 2021) and random cropping (Mi et al., 2021). Most importantly, CNNs can process arbitrary input sizes benefited by their inductive biases, *i.e.*, translation equivalence, and locality. For the trainable Twins in MVSFormer, training with the same resolution tends to overfit one input size, and fails to be generalized to HR cases. [2]

---

[2] For multi-crop augmented DINO frozen in MVSFormer-P, the scale issue is alleviated; but it still limits the performance (Tab. 4).

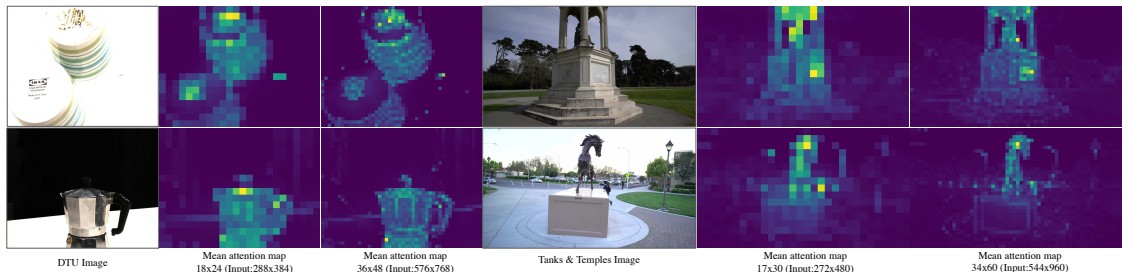

Figure 3: Mean attention maps of $[CLS]$ token from DINO (Caron et al., 2021) with various resolutions. Note that $576{\times}768$ and $544{\times}960$ are max input sizes for DTU and Tank-and-Temples to our MVSFormer-P.

Thus, we repurpose learning features with multi-scale training, originated from ViT-based detection tasks. Particularly, for efficient multi-scale training, we have to ensure that 1) image sizes for each batch should be the same; 2) dynamically changing the batch size according to image sizes, which aims to make the full usage of limited memory. Specifically, we train our models with dynamic resolutions from 512 to 1280, while the aspect ratios are randomly sampled from 0.8 to 0.67. Instead of the compromise by a minimum batch size, we keep the multi-scale training with the largest batch size, assisted by the gradient accumulation. The gradient accumulation splits a batch into several sub-batches and accumulates their gradients to update the model. All instances are grouped into different pairs of *resolution and sub-batch size* at the start of each epoch randomly. Note that a larger image should have a smaller sub-batch to balance the memory cost and vice versa. Training with a larger batch size contributes to faster convergence with lower variances and better performance for BatchNorm layers (Ioffe, 2017). Therefore, the gradient accumulation significantly improves the efficiency of multi-scale training of MVSFormer. We find that dynamic training sizes from 512 to 1280 are sufficient to generalize MVSFormer to at least 2K resolution of Tanks-and-Temples (Knapitsch et al., 2017) as compared in Tab. 16 and Fig. 11. More details about multi-scale training are in Sec. A.1 of Appendix.

### 3.3 Correlation Volume Construction

We elaborate the classical components of cost volume construction and regularization from Gu et al. (2020); Zhang et al. (2020); Giang et al. (2021). Note that such components are orthogonal to our key contributions, as the primary focus of this paper is on better feature extraction for MVS. To achieve the multi-stage cost volume, we first initialize a group of inverse depth ranges $\{d_j\}_{j=1}^{D}$ for each stage. Here we omit the superscript of $l$-th stage for the simplification. Features of source views are warped to the reference view. Given a 2D pixel $\mathbf{p}$ of the reference image $\mathbf{I}_0$ with known camera intrinsics $\mathbf{K}_0, \mathbf{K}_i$ of reference and source views, as well as their rotation $\mathbf{R}_{0 \to i}$ and translation $\mathbf{t}_{0 \to i}$, warped $\mathbf{p}'_j$ with the $j$-th depth hypothesis in source image $\mathbf{I}_i$ can be written as

$$\mathbf{p}'_j = \mathbf{K}_i \cdot (\mathbf{R}_{0 \to i} \cdot \mathbf{K}_0^{-1} \cdot \mathbf{p} \cdot d_j + \mathbf{t}_{0 \to i}). \tag{3}$$

Then the group-wise pooling (Guo et al., 2019) is leveraged to split features into $G$ groups along the channel dimension. And the feature correlation $\mathbf{C}_i$ can be formulated by the inner production of group-wise reference features $\mathbf{F}_0^g$ and warped source features $\mathbf{F}_i^g$ as

$$\mathbf{C}_i(d_j, \mathbf{p}, g) = \left\langle \mathbf{F}_0^g(\mathbf{p}), \mathbf{F}_i^g(\mathbf{p}'_j) \right\rangle \in \mathbb{R}^{\hat{C}}, \tag{4}$$

where $\hat{C}$ is the channel of $\mathbf{F}_0^g$ and $\mathbf{F}_i^g$. Then the feature correlation from Eq.(4) is further averaged for each group to $\mathbf{C}_i(d_j, \mathbf{p}) \in \mathbb{R}^G$ for an efficient cost volume formulation. We also train a 2D CNN to learn pixel-wise weight visibility $\{\mathbf{W}_i\}_{i=1}^{N-1}$ for each source view through the entropy of normalized correlations (Zhang et al., 2020; Giang et al., 2021). Thus $N-1$ source feature correlations can be fused with their visibility as

$$\mathbf{C}(d_j) = \frac{\sum_{i=1}^{N-1} \mathbf{W}_i \mathbf{C}_i(d_j)}{\sum_{i=1}^{N-1} \mathbf{W}_i}, \tag{5}$$

which is the input for the 3D U-Net cost volume regularization. After being regularized by the 3D U-Net, we can achieve pixel-wise output 3D cost volume $\hat{\mathbf{C}} \in \mathbb{R}^{D \times H_l \times W_l}$ for each stage.

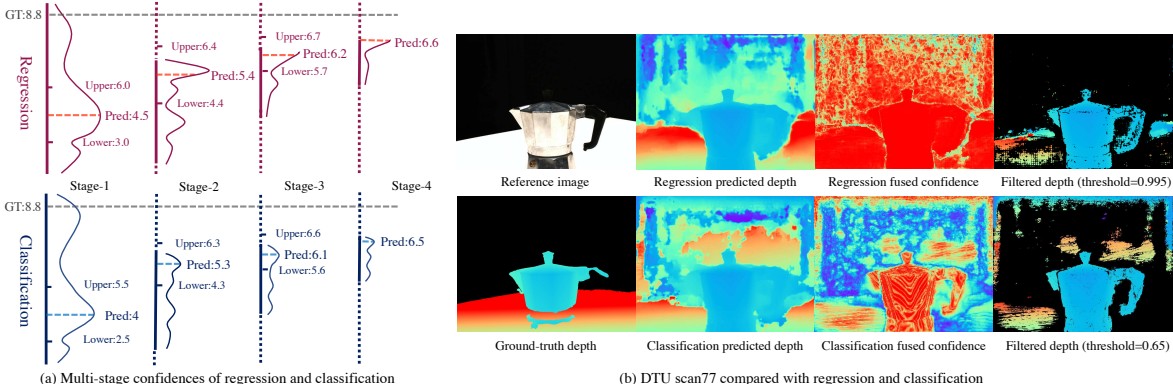

(a) Multi-stage confidences of regression and classification      (b) DTU scan77 compared with regression and classification

Figure 4: (a) Multi-stage confidence of regression and classification. Both depth ranges of regression and classification based methods miss ground truth depth (8.8) in stage-1. Regression still provides high probability for the depth upper bound. (b) An DTU example: the kettle handle can be kept by confidence map of classification-based method. Confidence maps are got by Eq.(8) of the Appendix.

## 3.4 Temperature-based Depth Prediction

Given the cost volume $\hat{\mathbf{C}}$ from the 3D U-Net, the probability volume of depth hypotheses can be achieved by the softmax along the depth dimension as $\mathbf{P} = \text{softmax}(\hat{\mathbf{C}})$. REGression-based depth (REG) utilizes *soft-argmin* (Kendall et al., 2017) to softly weighting for each depth hypothesis, *i.e.*, the expectation of $\{d_j\}_{j=1}^{D}$ with probability $\mathbf{P}(d_j)$. For the CLAssification-based depth (CLA), the predicted depth $\mathbf{D}_{cla}$ is simply selected from all depth hypotheses with the maximum probability, *i.e.* the *argmax* depth hypothesis. Thus the regressive depth $\mathbf{D}_{reg}$ and the classification depth $\mathbf{D}_{cla}$ are

$$\mathbf{D}_{reg} = \sum_{j=1}^{D} d_j \cdot \mathbf{P}(d_j), \quad \mathbf{D}_{cla} = \operatorname*{arg\,max}_{d_j \in \{d_j\}_{j=1}^{D}} \mathbf{P}(d_j). \tag{6}$$

$\mathbf{D}_{reg}$ is optimized with the $L_1$ loss with the ground-truth depth, while $\mathbf{D}_{cla}$ is optimized with the Cross-Entropy (CE) with the one-hot ground-truth depth volume.

**Remark**. Peng et al. (2022) think that REGs suffer from the overfitting issue, and lead to ambiguous depth prediction, whilst CLAs are more robust but fail to achieve exact depth results. In contrast, we empirically find a different conclusion that confidence maps from CLAs are better than the REGs; and this should not be neglected especially for the widely used multi-stage MVS models (Gu et al., 2020; Yang et al., 2020; Mi et al., 2021). Particularly, MVS networks can not ensure all predicted depth maps are correct, due to reflection, occlusion, or missing reliable source views. Thus, providing solid confidence (uncertainty) maps of depth (Kendall & Gal, 2017) is also important for MVS to reconstruct good point clouds as empirically evaluated in Fig. 6 and Fig. 13. The way to get confidence maps of REGs and CLAs are summarized in Sec. A.4. However, as shown in Fig. 4(a), the REG maintains high-confidence values even for out-of-range depth hypotheses in stage-2,3,4. It is difficult for REGs to filter outliers without hurting other points of correct depth as illustrated in Fig. 4(b). Since CE can not tackle out-of-range depth labels, we advocate masking all depth outliers in training as Mi et al. (2021). Note that we also tried to optimize the MVS with masked $L_1$ loss, but its performance is inferior to regular regression.

Although CLA has many good properties for MVS, REGs can achieve superior performance of depth and point cloud compared with CLAs in our early experiments. So we target on the issue mentioned in Peng et al. (2022), *i.e.*, inexact depth predictions. UniMVSNet (Peng et al., 2022) designs a Unified Focal Loss (UFL) to solve it, which regards CE as multiple Binary Cross-Entropy (BCE). And the focal loss (Lin et al., 2017b) controlled by several hyper-parameters is used to solve the imbalance problem in BCE. Different from UFL, we propose a simple way to unify both REGs and CLAs, which only adjusts the inference process without re-training the model. We first multiply a temperature $t$ to the cost volume $\hat{\mathbf{C}}$ before the softmax,

Table 1: Quantitative point cloud results (mm) on DTU (lower is better). Best results are in bold, and second ones are underlined. * denotes that GBiNet is re-tested with the same post-processing threshold to all scans for fair comparisons with other methods.

| Methods | Accuracy↓ | Completeness ↓ | Overall↓ |
|---|---|---|---|
| Gipuma (Galliani et al., 2015) | **0.283** | 0.873 | 0.578 |
| COLMAP (Schönberger et al., 2016) | 0.400 | 0.664 | 0.532 |
| R-MVSNet (Yao et al., 2019) | 0.385 | 0.459 | 0.422 |
| AA-RMVSNet (Wei et al., 2021) | 0.376 | 0.339 | 0.357 |
| CasMVSNet (Gu et al., 2020) | 0.325 | 0.385 | 0.355 |
| CDS-MVSNet (Giang et al., 2021) | 0.352 | 0.280 | 0.316 |
| UniMVSNet (Peng et al., 2022) | 0.352 | 0.278 | 0.315 |
| TransMVSNet (Ding et al., 2021) | 0.321 | 0.289 | 0.305 |
| GBiNet* (Mi et al., 2021) | 0.312 | 0.293 | 0.303 |
| MVSFormer (ours) | 0.327 | **0.251** | **0.289** |
| MVSFormer-P (ours) | 0.327 | 0.265 | 0.296 |

and rewrite $\mathbf{D}_{reg}$ to the temperature-based depth expectation $\mathbf{D}_{tmp}$ as

$$\mathbf{D}_{tmp} = \sum_{j=1}^{D} d_j \cdot \hat{\mathbf{P}}(d_j), \quad \hat{\mathbf{P}} = \text{softmax}(\hat{\mathbf{C}} \cdot t). \tag{7}$$

Obviously, when $t = \infty$ or $t = 1$, $\mathbf{D}_{tmp}$ is equivalent to $\mathbf{D}_{cla}$, or $\mathbf{D}_{reg}$, respectively. The core idea is to adjust the temperature $t$ during the inference to unify CLAs and REGs. For early stages with low-resolution, we set larger $t$ to make the model work as a CLA for a better global distinguishing ability. And for later stages with high-resolution, our model tends to use lower $t$ as a REG to smooth local details. In practice, we set $\{t^1, t^2, t^3, t^4\} = \{5, 2.5, 1.5, 1\}^3$ and achieve better performance than classification ($t = \infty$), regression ($t = 1$), and other consistent settings of $t$ as evaluated in Sec. 4.2. Note that $\mathbf{D}_{tmp}$ is only used during testing, as the masked CLA optimized with CE is robust enough for MVS learning. Thus, $\mathbf{D}_{cla}$ is adopted in MVSFormer for the training phase. Although adjusting the temperature during the testing may suffer from some implications of the discrepancy between the train and test stages, we just tend to regress in the latter stages with only a few nearby depth hypotheses. Thus such gap is largely narrowed. And our temperature setting is generalized and effective enough for various datasets. More details are discussed in Sec. C.6.

## 4 Experiments

**Settings.** Our methods are evaluated on DTU (Aanæs et al., 2016), Tanks-and-Temples (Knapitsch et al., 2017) and ETH3D (Schops et al., 2017). Since DTU data is collected in an indoor environment with fixed camera poses, our model is finetuned on the BlendedMVS dataset (Yao et al., 2020) with various scenes and objects to generalize more complex environments in Tanks-and-Temples and ETH3D, as standard practice in Giang et al. (2021); Ding et al. (2021). MVSFormer is trained by the view number $N = 5$ of 4 coarse-to-fine stages of 32-16-8-4 depth hypotheses. CNN parts in MVSFormer are trained by Adam with a learning rate of 1e-3. The part of Twins-small in MVSFormer is trained with learning rate 3e-5 and 0.01 weights decay, while DINO-small is frozen in MVSFormer-P. Our models are trained by 10 epochs on DTU and finetuned with another 10 epochs on BlendedMVS. The learning rate is warmed up with 500 steps and then decayed with the cosine scheduler. For the multi-scale training, we dynamically change the sub-batch from 8 to 2 according to the scales from 512 to 1280, with a maximum batch size of 8. More details are in Appendix Sec. A.

### 4.1 Quantitative Results

**DTU.** Our MVSFormer is evaluated on DTU with the official evaluation metrics of point clouds, *i.e,* accuracy, completeness, and the overall error. The testing resolution is fixed in $1152 \times 1536$ and the view number

---

[3]Such a temperature setting is always fixed for all datasets and instances without data dependency. As analyzed in Sec. C.6, this setting may not be the best one, but the difference is not obvious. And the critical idea of 'classify first, then regress' is the same.

Table 2: Quantitative results of F-score on Tanks-and-Temples. Higher F-score means a better reconstruction quality. Best results are in bold, while the second ones are underlined.

| Methods | Intermediate | | | | | | | | | Advanced | | | | | | |
|---|---|---|---|---|---|---|---|---|---|---|---|---|---|---|---|---|
| | Mean | Fam. | Fra. | Hor. | Lig. | M60 | Pan. | Pla. | Tra. | Mean | Aud. | Bal. | Cou. | Mus. | Pal. | Tem. |
| COLMAP | 42.14 | 50.41 | 22.25 | 26.63 | 56.43 | 44.83 | 46.97 | 48.53 | 42.04 | 27.24 | 16.02 | 25.23 | 34.70 | 41.51 | 18.05 | 27.94 |
| CasMVSNet | 56.84 | 76.37 | 58.45 | 46.26 | 55.81 | 56.11 | 54.06 | 58.18 | 49.51 | 31.12 | 19.81 | 38.46 | 29.10 | 43.87 | 27.36 | 28.11 |
| CDS-MVSNet | 61.58 | 78.85 | 63.17 | 53.04 | 61.34 | 62.63 | 59.06 | 62.28 | 52.30 | – | – | – | – | – | – | – |
| TransMVSNet | 63.52 | 80.92 | 65.83 | 56.94 | 62.54 | 63.06 | 60.00 | 60.20 | 58.67 | 37.00 | 24.84 | 44.59 | 34.77 | 46.49 | 34.69 | 36.62 |
| UniMVSNet | 64.36 | 81.20 | 66.43 | 53.11 | 63.46 | **66.09** | **64.84** | **62.23** | 57.53 | 38.96 | 28.33 | 44.36 | **39.74** | **52.89** | 33.80 | 34.63 |
| GBiNet | 61.42 | 79.77 | 67.69 | 51.81 | 61.25 | 60.37 | 55.87 | 60.67 | 53.89 | 37.32 | **29.77** | 42.12 | 36.30 | 47.69 | 31.11 | 36.93 |
| **MVSFormer** | **66.37** | **82.06** | **69.34** | **60.49** | **68.61** | 65.67 | 64.08 | 61.23 | **59.53** | **40.87** | 28.22 | **46.75** | 39.30 | 52.88 | **35.16** | **42.95** |

$N = 5$. We use the depth fusion of Gipuma (Galliani et al., 2015) with a consistent confidence threshold 0.5 for the point clouds. Quantitative results of DTU are shown in Tab. 1, and qualitative ones are shown in Fig. 9 and Fig. 10 of the Appendix. Note that the post-processing hyper-parameters of all scans are fixed for learning-based methods. Traditional methods (Galliani et al., 2015; Schönberger et al., 2016) fail to get good completeness, which means that they have missed many points with sparse results. For learning-based methods, Our full trainable MVSFormer can achieve the best completeness and overall error. MVSFormer-P is the second-best in overall error which enjoys faster efficiency. Therefore, our methods can get more complete point clouds compared with other competitors. Note that results reported in GBiNet (Mi et al., 2021) need to use different hyper-parameters for the post-processing. Our methods can outperform GBiNet with fixed hyper-parameter settings for all scans. With the impressive improvements achieved by MVSFormer, we think that pre-trained ViTs have the potential to push the limits of MVS.

**Tanks-and-Temples.** Our submission of the full trainable MVSFormer has *ranked Top-1 on both intermediate and advanced sets of the official Tanks-and-Temples leaderboard* compared with other published works since May/2022. We show quantitative results on both intermediate and advanced sets in Tab. 16. All instances are inferred with the original $1088 \times 1920$ image size, and more detailed settings are in the Appendix. The metric is officially evaluated by the F-score based on precision and recall of submitted point clouds (Knapitsch et al., 2017). MVSFormer outperforms all other state-of-the-art methods with mean F-scores of 66.37 and 40.87 for intermediate and advanced sets respectively. As shown in Tab. 16, our method can achieve the best or second-best results in almost all cases except 'Auditorium', which demonstrates its good generalization and impressive performance. Furthermore, confidence maps from CLA can filter outliers and get more precise point clouds as shown in the Appendix. Besides, the proposed multi-scale strategy can generalize MVSFormer to fit larger resolutions, such as 2K. More qualitative results are in Fig. 11 and Fig. 12 of the Appendix.

**ETH3D.** To show the robustness of the proposed method in scene data, we additionally evaluate MVSFormer on both training and test set of the high-resolution ETH3D (Schops et al., 2017) without re-training. ETH3D contains 13 training and 12 test scenes, which include both indoor and outdoor challenging scenes. We evaluate the ETH3D with the model trained on DTU and finetuned on BlendedMVS. Input images are resized into $1088 \times 1920$ as Tanks-and-Temples and the view number is 7. Other settings are the same as Wang et al. (2022a). MVSFormer is quantitatively compared with both traditional methods (Galliani et al., 2015; Schönberger et al., 2016) and other state-of-the-art learning-based ones (Wang et al., 2021a; 2022a;b) in Tab. 3. MVSFormer achieves the best F1-score in both the training and test set of ETH3D, which shows good robustness and generalization of our method on scene datasets.

## 4.2 Ablation studies

**Different pre-trained models.** We have tested different pre-trained models for MVS in Tab. 4, which include ResNet50 (He et al., 2016), DINO (Caron et al., 2021), MAE (He et al., 2021), and Twins (Chu et al., 2021a). Note that both ResNet50 and Twins are trainable, while DINO and MAE are frozen. Our baseline method is a 4-stage cascaded MVS model with visibility modules (Zhang et al., 2020) and the random cropping (Mi et al., 2021). From Tab. 4, ResNet50 in row2 improves the depth but fails to reduce the overall error of point clouds. Because CNN-based pre-training can not learn proper features from reflection and

Table 3: Comparisons of Accuracy (Acc.), Completeness (Cop.) and F1-score (F1) on the ETH3D benchmark.

| Methods | Training set | | | Test set | | |
|---|---|---|---|---|---|---|
| | Acc. | Cop. | F1 | Acc. | Cop. | F1 |
| Gipuma (Galliani et al., 2015) | 84.44 | 34.91 | 36.38 | 86.47 | 24.91 | 45.18 |
| PMVS (Furukawa & Ponce, 2009) | 90.23 | 32.08 | 46.06 | 90.08 | 31.84 | 44.16 |
| COLMAP (Schönberger et al., 2016) | **91.85** | 55.13 | 67.66 | **91.97** | 62.98 | 73.01 |
| ACMH (Xu & Tao, 2019) | 88.94 | 61.59 | 70.71 | 89.34 | 68.62 | 75.89 |
| PatchMatchNet (Wang et al., 2021a) | 64.81 | 65.43 | 64.21 | 69.71 | 77.46 | 73.12 |
| PatchMatch-RL (Lee et al., 2021) | 76.05 | 62.22 | 67.78 | 74.48 | 72.89 | 72.38 |
| CDS-MVSNet (Giang et al., 2021) | 73.07 | 64.22 | 67.65 | 77.32 | 81.43 | 79.07 |
| MVSTER (Wang et al., 2022b) | 76.92 | 68.08 | 72.06 | 77.09 | 82.47 | 79.01 |
| IterMVS (Wang et al., 2022a) | 79.79 | 66.08 | 71.69 | 84.73 | 76.49 | 80.06 |
| MVSFormer (ours) | 73.62 | **74.64** | **73.44** | 82.23 | **83.75** | **82.85** |

Table 4: Ablations in DTU on different pre-trained models, augmentation strategies, and loss types. Metrics are depth error ratios of 2mm ($e_2$), 4mm ($e_4$), 8mm ($e_8$) and Overall error (Ovl.) of point clouds. Red line denotes our baseline results; green and blue rows refer to our MVSFormer-P and MVSFormer respectively.

| Pre-trained | Augmentation | | | Loss | | $e_2 \downarrow$ | $e_4 \downarrow$ | $e_8 \downarrow$ | Ovl.$\downarrow$ |
|---|---|---|---|---|---|---|---|---|---|
| | Cropping | HR-FT | Multi-scale | REG | CLA | | | | |
| – | ✓ | | | ✓ | | 22.81 | 18.12 | 14.85 | 0.321 |
| ResNet50 | ✓ | | | ✓ | | 20.09 | 15.11 | 11.78 | 0.323 |
| ResNet50 | | | ✓ | | ✓ | 20.38 | 13.87 | 10.37 | 0.312 |
| DINO-small | ✓ | | | ✓ | | 22.06 | 16.63 | 12.78 | 0.309 |
| DINO-small | | | ✓ | ✓ | | **17.06** | **11.60** | **7.97** | 0.301 |
| DINO-small | ✓ | ✓ | | ✓ | | 20.98 | 15.13 | 10.73 | 0.309 |
| DINO-small | | | ✓ | | ✓ | 17.18 | 11.96 | 8.53 | 0.296 |
| MAE-base | | | ✓ | ✓ | | 18.45 | 13.07 | 9.36 | 0.307 |
| Twins-small | ✓ | | | ✓ | | 22.62 | 17.51 | 13.75 | 0.312 |
| Twins-small | | | ✓ | ✓ | | 17.70 | 12.47 | 8.92 | 0.293 |
| Twins-small | | | ✓ | | ✓ | 17.50 | 12.48 | 9.14 | **0.289** |

texture-less areas, which causes discouraged metrics for these scans and leads to even worse results in point clouds as shown in Fig. 5. Our MVSFormers (row4 and row9 of Tab. 4) can improve the baseline even without the multi-scale training. For the comparisons among frozen pre-trained ViTs with the multi-scale training strategy, DINO-small achieves better performance with fewer parameters compared with MAE-base, which is benefited by the multi-crop (Caron et al., 2020) and the proposed attention based GLU. Although REG based methods enjoy slightly better depth, they fail to reconstruct proper point clouds with proper confidence. The Twins and DINO based MVSFormers achieve the best results in point cloud and depth respectively, while the pre-trained ResNet50 based model with all other tricks (row3 of Tab. 4) fails to produce competitive results. Detailed comparison among MVSFormers and ResNet are discussed in Sec. C.5. The computation and GPU memory cost of different pre-trained MVS methods are discussed in Sec. B.

**Multi-scale training.** From Tab. 4, both Twins and DINO based MVSFormers achieve considerable improvements from the multi-scale training. Especially, the trainable Twins enjoys much more benefits (0.19

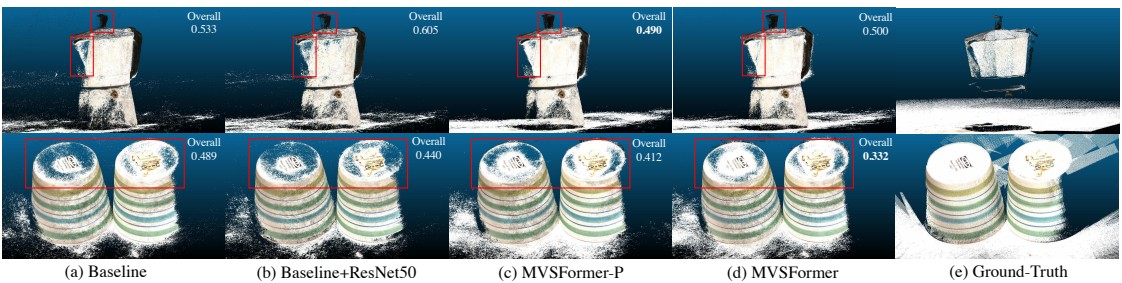

(a) Baseline  (b) Baseline+ResNet50  (c) MVSFormer-P  (d) MVSFormer  (e) Ground-Truth

Figure 5: Qualitative results compared with our baseline worked with different pre-trained models.

Table 5: Left table: Ablation study about pre-training of Twins in MVSFormer. Right table: The ablation study of DINO attention map and GLU in MVSFormer-P. 'MS' means the multi-scale training. Blue and green rows show the performance of the full model of MVSFormer and MVSFormer-P respectively.

| Pre-trained | $e_2\downarrow$ | $e_4\downarrow$ | $e_8\downarrow$ | Ovl.$\downarrow$ |
|---|---|---|---|---|
| × | 20.16 | 14.91 | 11.02 | 0.300 |
| ✓ | **17.50** | **12.48** | **9.14** | **0.289** |

| GLU | Concat+2Conv | MS+CLA | $e_2\downarrow$ | $e_4\downarrow$ | $e_8\downarrow$ | Ovl.$\downarrow$ |
|---|---|---|---|---|---|---|
| – | – | | 24.12 | 18.29 | 13.66 | 0.312 |
| | ✓ | | 22.78 | 17.25 | 13.30 | 0.310 |
| ✓ | | | 22.06 | 16.63 | 12.78 | 0.309 |
| ✓ | | ✓ | **17.18** | **11.96** | **8.53** | **0.296** |

Table 6: Depth ($e_2$, $e_4$, $e_8$) and point cloud ablations of the Twins based MVSFormer trained with REG and CLA, while the CLA model is further inferenced with different temperatures $t$.

| REG | CLA | $t$ | $e_2\downarrow$ | $e_4\downarrow$ | $e_8\downarrow$ | Accuracy$\downarrow$ | Completeness$\downarrow$ | Overall$\downarrow$ |
|---|---|---|---|---|---|---|---|---|
| ✓ | | – | 17.70 | 12.47 | **8.92** | 0.336 | 0.249 | 0.293 |
| | ✓ | $\infty$ | 18.03 | **12.32** | 8.94 | 0.349 | 0.248 | 0.298 |
| | ✓ | 3.0 | 17.62 | 12.48 | 9.18 | 0.342 | **0.247** | 0.295 |
| | ✓ | 2.0 | 17.78 | 12.72 | 9.37 | 0.336 | 0.250 | 0.293 |
| | ✓ | 1.0 | 19.06 | 13.97 | 10.37 | 0.323 | 0.258 | 0.291 |
| | ✓ | 0.75 | 19.99 | 14.88 | 11.11 | **0.321** | 0.270 | 0.296 |
| | ✓ | $\{5, 2.5, 1.5, 1\}$ | **17.50** | 12.48 | 9.14 | 0.327 | 0.251 | **0.289** |

in the overall error). Because of the spatial invariance, CNN-based ResNet50 basically can not be benefited from multi-scale training (row3). Besides, we also compare multi-scale strategy with high-resolution finetuning (HR-FT), *i.e.*, finetuning the model with fixed 1024×1280 images for another 5 epochs. But HR-FT produce inferior results compared with the multi-scale one. We think that finetuning on a specific resolution still suffers from the spatial overfitting to attention blocks, which limits the generalization of ViTs for various image scales.

**The effect of pre-training for ViTs.** Although training a transformer (with CNN) for MVS is feasible (Zhu et al., 2021), we think that the pre-training is still important for MVS learning, especially for the feature learning in our work. In particular, we train our MVSFormer with Twins-small from the scratch as shown in Tab. 5(left). We increase the learning rate of no pre-trained Twins-small from 3e-5 to 1e-4, while all other settings are unchanged. Results from Tab. 5(left) show that pre-training is critical to our proposed MVSFormer. The Twins-small without pre-training is not as good as pre-trained one. Without the pre-training, our methods are close to those attention-based MVS methods (Ding et al., 2021; Zhu et al., 2021) with only intra-view attention except the temperature based depth. Actually, no pre-trained MVSFormer performs similarly as TransMVSNet (Ding et al., 2021). So the pre-training is important for ViTs to model proper feature representations to tackle the essential feature matching problem in MVS.

**Effects of DINO attention maps from [*CLS*] and GLU.** In Tab. 5(right), we evaluate the performance of MVSFormer-P with and without the [*CLS*] attention map. Furthermore, we compare GLU based attention map fusion with the simple concatenation and ×2 convolutions to balance the parameters. From Tab. 5(right), both attention fusion strategies enjoy improvements compared with the baseline without such attention map. And the GLU block can achieve better depth with the same computation.

**REG vs CLA in MVS.** To show the importance of effect confidence map from CLA models during point cloud reconstruction. In Fig. 6, we further show the difference between the REG and CLA on the real-world Tanks-and-Temples reconstruction. Note that CLA can produce more reliable confidence maps to distinguish certain foregrounds and uncertain backgrounds (*e.g.*, sky). Thus CLA based methods reconstruct more valid points with much less outliers, which is crucial for the real-world MVS practice. More real-world cases are shown in Fig. 13 of Appendix.

**Temperature-based depth prediction.** Quantitative ablations about REG and CLA of MVSFormer are shown in Tab. 6. The vanilla CLA ($t = \infty$) based model can not achieve better results compared with the one trained with REG. As reducing the temperature $t$, depth is smoothed and models tend to reconstruct more accurate point clouds. Because of the decrease in accuracy distance and the increase in completeness

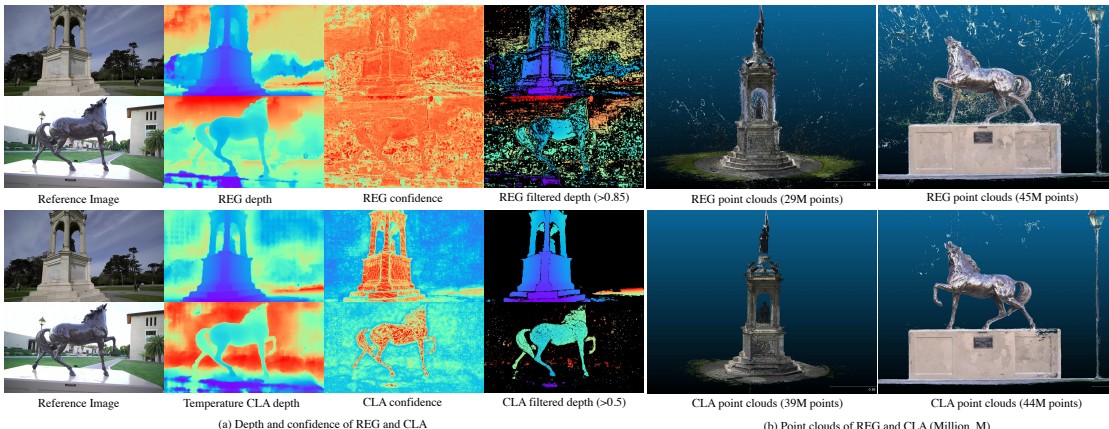

Figure 6: 'Francis' and 'Horse' in Tanks-and-Temples compared between REG and CLA based MVSFormer. The number of total points are listed in the brackets of (b).

distance. Since the trade-off of the accuracy and completeness, $t = 0.75$ achieves a worse overall result. Simply reducing $t$ for early stages causes over-smoothing results, which harms the completeness of point clouds. Our $\{t^1, t^2, t^3, t^4\} = \{5, 2.5, 1.5, 1\}$ setting can get a good trade-off in both depth and point clouds, which is better than any consistent $t$; it also enjoys better accuracy compared with REG, which indicates that CLA provides better confidence maps to filter outliers. Therefore, the idea of making early stages work as CLA and latter stages work as REG is reasonable. More qualitative (Sec. C.3) and quantitative (Sec. C.6) analysis about the temperature are discussed in the Appendix.

## 5 Conclusion

In this paper, we discuss the influence of pre-trained models on MVS learning, and propose a ViT enhanced MVS architecture called MVSFormer. MVSFormer achieves prominent improvements with a better feature encoding enhanced by pre-trained hierarchical-ViT Twins. Furthermore, we propose an alternative MVSFormer-P with the plain-ViT DINO, which can also achieve competitive results with a frozen backbone. The efficient multi-scale training is used to generalize MVSFormer to various resolutions. Besides, an effective temperature-based depth prediction is proposed to unify both REG and CLA in the MVS learning, which produces smooth depth maps and clear point clouds without outliers. Our method can achieve state-of-the-art results in DTU, and rank top-1 on Tanks-and-Temples.

### Acknowledgments

This work was supported by NSFC under Grant (No.62076067).

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

# A More Implementation Details and Discussions

## A.1 Multi-scale Training

The PyTorch pseudo-code of the multi-scale training is summarized in Alg. 1. The training scales are randomly selected from 25 patterns, whose height is ranged from 512 to 1024, while width is ranged from 640 to 1280. We also randomly crop images from [0.83,1.0] of the original scale. The global batch size is 8, while the relation of sub-batch size and resolution is shown in Tab. 7. Thanks for the mixed-precision, it only takes about 22 and 15 hours for our proposed MVSFormer to be trained with 10 epochs in DTU (Aanæs et al., 2016) and BlendedMVS (Yao et al., 2020) respectively with two V100 32GB NVIDIA Tesla GPUs.

---

**Algorithm 1** PyTorch pseudo code for efficient multi-scale training with gradient accumulation

```
# B: maximal batch size
# scale_batch_dict: a mapping dict, key is resolution; value is related sub-batch size
# optimizer: Adam optimizer
for (x, y) in data_loader: # load a batch consisted of inputs x and reference depth y
    b = scale_batch_dict[shape(x)] # get related sub-batch size from the dict
    n = B // b # get the step number n for accumulation
    for i in range(n): # accumulate gradients for n steps
        y_pred = MVSFormer(x[i*b:(i+1)*b]) # model inference with a sub-batch
        loss = LossFunction(y_pred, y[i*b:(i+1)*b]) # calculate loss for sub-batch
        loss.backward() # back-propagate and accumulate gradients
    optimizer.step() # optimize model parameters
    optimizer.zero_grad() # clear accumulated gradients
```

---

Table 7: The relation of sub-batch size and resolution in the multi-scale training of MVSFormer-P and MVSFormer, which can be trained in two 32GB GPUs with maximum batch size 8.

| Resolution | sub-batch size | |
| --- | --- | --- |
| | MVSFormer-P | MVSFormer |
| 512x640~768 | 8 | 8 |
| 576x704~832 | 8 | 8 |
| 640x832~960 | 8 | 8 |
| 704x896~1024 | 8 | 4 |
| 768x960~1088 | 4 | 4 |
| 896x1152~1280 | 4 | 4 |
| 960x1216~1344 | 4 | 2 |
| 1024x1280 | 4 | 2 |

## A.2 Data Augmentation

We augment input images for ViT in MVSFormer with the same parameter for all reference and source views, which can slightly improve the depth performance as shown in Tab. 8. Specifically, the data augmentation is performed by random gamma, color jitters with brightness, contrast, saturation, and hue. The augmented parameters are sampled from uniform distributions in the ranges [0.9,1.1] for gamma, contrast and saturation, [0.8,1.2] for brightness, [0.95,1.05] for hue.

Table 8: Depth metrics on DTU ($512 \times 640$) with error ratios of 2mm ($e_2$), 4mm ($e_4$), 8mm ($e_8$), and 14mm ($e_{14}$) with/without data augmentation.

|  | $e_2 \downarrow$ | $e_4 \downarrow$ | $e_8 \downarrow$ | $e_{14} \downarrow$ |
|---|---|---|---|---|
| w./o. aug | 17.15 | 10.85 | 7.29 | 5.38 |
| with aug | **16.52** | **10.46** | **7.07** | **5.22** |

## A.3 Post-processing

For DTU, we get the final point clouds with the depth fusion tool from Gipuma (Galliani et al., 2015) with consistent hyper-parameters, *i.e.*, disparity threshold 0.1, number consistent 2, and probability threshold 0.5. For Tanks-and-Temples, to avoid adjusting hyper-parameters for each case, we follow the dynamic consistency checking proposed in Yan et al. (2020).

## A.4 Confidence Maps

We achieve confidence maps for REGs and CLAs of each stage as

$$\mathbf{P}_{reg} = \text{AvgPool}(\mathbf{P}, k)(\mathbf{D}_{reg}), \quad \mathbf{P}_{cla} = \mathbf{P}(\mathbf{D}_{cla}), \tag{8}$$

where $\mathbf{D}_{reg}$ and $\mathbf{D}_{cla}$ are predicted depth from Eq. 6; $\mathbf{P}$ is the probability after the softmax along depth hypotheses; $k$ indicates the kernel size for the depth pooling. For CLAs we simply use the max probability of all depth hypotheses as the confidence map. REGs follow Yao et al. (2018) that averages the probability along the depth before gathering the confidence. For the cascaded depth hypotheses, we set the pooling kernel size $k$ for 4 stages as $\{4,3,2,1\}$ to 32-16-8-4 hypotheses instead of fixing in 4 of Yao et al. (2018) for slightly better reconstruction results. Then confidence maps of all stages are averaged for the final confidence output after the nearest resizing to the original image size.

## A.5 Additional Discussions

**Why completely separate the FPN and ViT in MVSFormer?** The FPN used in MVS can efficiently integrate the information from different input scales. And FPN is not redundant to our ViT in MVSFormer for three reasons. 1) We use 1/2 input scale for ViT, while FPN takes full resolution inputs. So FPN can learn more detailed information. In Sec. C.1, we have discussed the performance of ViTs without FPNs and pre-trained ResNet. ViTs can work complementarily with FPNs for both global understanding and local details in MVS. 2) Since the inputs to ViT are downsampled to 1/2 to balance the computation, the valid largest feature scale of Hierarchical ViT is 1/8, while the one of Plain ViT is smaller (1/32). Our feature scales of FPN are 1, 1/2, 1/4, 1/8 respectively. So only one feature layer (1/8) can be completely replaced with the ViT feature. For the reason of the model integrity, we remain all layers of the FPN. 3) Because the inputs to FPN are 2x larger than ones to ViT, the highest level feature (1/8) of FPN also contains informative features learned from high-resolution inputs which are not included in ViTs.

**Why using DINO attention maps from [$CLS$] rather than other tokens?** We use the attention map attached from the [$CLS$] token for two reasons. 1) DINO (Caron et al., 2021) is pre-trained with self-distillation. The [$CLS$] token is directly supervised during the pre-training. So the [$CLS$] token needs to understand the global semantics in the image. Thus the attention map of the [$CLS$] token is more significant to decide the final prediction. 2) Moreover, the attention map from all tokens ($HW \times HW$) is much larger

than ones from $[CLS]$ token ($H \times W$). Thus choosing or getting a reasonable attention map from $HW \times HW$ relations is also a difficult issue.

**Relation of CLA/REG confidence and aleatoric/epistemic uncertainty.** As mentioned in Kendall & Gal (2017), aleatoric uncertainty captures noise inherent in the observations, while epistemic uncertainty indicates the uncertainty in the model parameters. In terms of the definitions from Kendall & Gal (2017), both CLA and REG should be categorized as aleatoric uncertainty rather than epistemic uncertainty. Because we have not tested these methods with model uncertainty (*e.g.*, dropout inference). More importantly, both CLA and REG only show the uncertainty of noise in the training distribution, rather than the Out-of-data examples. For example, most learning based MVS methods (both CLA and REG) demand a certain depth range before the prediction; and they cannot provide proper results with wrong depth ranges. However, given a certain depth range, CLA based cascade MVS enjoys much more reasonable confidence as discussed in Sec. 3.4. Moreover, the uncertainty estimation of MVS is an interesting issue as the future work.

## B    Inference Memory and Time Costs

We test the inference memory and time costs with input resolution $1152 \times 1536$ in Tab. 9 compared with CNN-based pre-training, pure FPN, and other MVS methods. All comparisons are based on a V100 NVIDIA Tesla GPU. From Tab. 9, although the parameter scale is increased, ViT enhanced MVSFormers only cost a little more GPU memory and time compared with the baseline method. Except for parameters contained by ViT itself, most other trainable parameters are worked for the dimension reduction in MVSFormer. MVSFormer can infer faster compared with MVSFormer-P, which is benefited from the efficient multi-scale attention designed in Twins (Chu et al., 2021a). Note that the pre-trained CNN model –ResNet50 also costs a lot in GPU memory and inference time. For other MVS algorithms, they are all forwarded with 1 view at once. CasMVSNet (Gu et al., 2020) costs most memory without the group-wise pooling (Guo et al., 2019) for the cost volume. The sophisticated TransMVSNet (Ding et al., 2021) suffers from slow inference speed and large memory costs. CDS-MVSNet (Giang et al., 2021) is slightly more efficient than MVSFormer, but it is discouraged in its performance.

Table 9: Illustration of model parameters (Params.) and memory/time-cost during the inference phase of $1152 \times 1536$ images. Results in brackets are forwarded with 5 views at once, while ones outside are forwarded 1 view at once. We also provide the efficiency of other MVS methods with their official codes (1-view).

|  | Memory (MB) | Time (s/img) | Params. (all) | Params. (trainable) |
|---|---|---|---|---|
| Baseline | 4262(8997) | 0.4125(0.4055) | 1.35M | 1.35M |
| Baseline+ResNet50 | 4882(9523) | 0.4920(0.4704) | 37.27M | 37.27M |
| MVSFormer-P | 4842(9251) | 0.5530(0.5230) | 26.45M | 4.79M |
| MVSFormer | 4970(9431) | 0.4831(0.4398) | 28.01M | 28.01M |
| CasMVSNet | 6672 | 0.4747 | 0.93M | 0.93M |
| CDS-MVSNet | 4760 | 0.4006 | 0.98M | 0.98M |
| TransMVSNet | 6320 | 1.4975 | 1.15M | 1.15M |

## C    More Experiment Results

### C.1    Pre-trained ViTs without FPNs

To explore the learning ability of ViTs, some quantitative results about ViTs (DINO-small (Caron et al., 2021), MAE-base (He et al., 2021), Twins-small (Chu et al., 2021a)) trained without FPNs for MVS are shown in Tab. 10, which are also compared with CNN-based pre-trained ResNet34 and ResNet50 and the vanilla FPN. Note that these experiments are only based on low-resolution cases in DTU of $256 \times 320$, because we want to save computation with all trainable ViT weights. The learning rates of MAE and DINO are 1e-5; the learning rate of Twins is 3e-5, and the ones of all CNNs are 1e-3. From Tab. 10, ViTs cannot achieve as good details as CNNs (2mm, 4mm), but results from ViTs are more robust in large depth error metrics (8mm,

14mm). Therefore, we think that ViTs can tackle some serious mistakes caused by reflection and texture-less areas as mentioned in the main paper. Interestingly, FPN trained from scratch achieves better depth results in low-resolution cases compared with other pre-trained CNNs, which demonstrates the dilemma of using pre-trained CNNs in MVS again. Twins-small can get better depth than FPN except for the 2mm error benefited by its pyramid architecture. So ViTs can work complementarily with CNN-based FPNs for both global understanding and local details in MVS.

Table 10: Depth metrics for DTU ($256 \times 320$) compared with FPN, pre-trained CNNs and ViTs (He et al., 2021; Caron et al., 2021; Chu et al., 2021a). Note that FPN is trained from scratch without pre-training.

|  |  | FPN | ResNet34 | ResNet50 | MAE-base | DINO-small | Twins-small |
|---|---|---|---|---|---|---|---|
| Error(%)$\rightarrow$ | 2mm | **19.53** | 20.65 | 20.55 | 21.85 | 26.66 | 19.65 |
| | 4mm | 11.85 | 12.43 | 12.32 | 12.42 | 14.42 | **11.53** |
| | 8mm | 8.25 | 8.55 | 8.45 | 8.17 | 8.87 | **7.92** |
| | 14mm | 6.48 | 6.75 | 6.60 | 6.33 | 6.60 | **6.24** |
| | mean | 11.52 | 12.09 | 11.98 | 12.19 | 14.14 | **11.34** |

### C.2   Different Feature Fusion Strategies of MVSFormer

We pay more attention to the essential improvements gained from pre-trained ViTs in this paper. Thus we tend to use simple feature fusion strategies in MVSFormer. Both the Direct Feature Addition (DFA) (used in the main paper) and the Multi-scale Feature Addition (MFA) is considered in Tab. 11. For the multi-scale addition, we use extra convolution blocks to further upsample ViT features to 1/4 and 1/2, and add them to related feature maps in FPN. Since inputs to ViTs are halved, we do not try to upsample ViT features to 1/1. From Tab. 11, MFA can achieve slightly better depth predictions but worse point cloud metrics. We think that ViT features are not suitable for high-resolution features, and adopt DFA as our solution.

Table 11: Ablations about different feature fusion strategies of Direct Feature Addition (DFA) and Multi-scale Feature Addition (MFA) in MVSFormer.

|  | $e_2 \downarrow$ | $e_4 \downarrow$ | $e_8 \downarrow$ | Acc.$\downarrow$ | Cop.$\downarrow$ | Ovl.$\downarrow$ |
|---|---|---|---|---|---|---|
| DFA | **17.50** | 12.48 | 9.14 | **0.327** | **0.251** | **0.289** |
| MFA | 17.53 | **12.24** | **8.62** | 0.329 | 0.253 | 0.291 |

### C.3   Qualitative Results of Temperature-based Depth Prediction

We show additional qualitative results of CLA with different temperature $t$ in Fig. 7. $t = 100$ tends to output depth maps with jagged boundaries, while depth maps with $t = 1$ and based on regression suffer from uncertain and ambiguous predictions. Although the visual difference is not obvious between Fig. 7(e) and Fig. 7(f), our setting of $\{t_1, t_2, t_3, t_4 = 5, 2.5, 1.5, 1\}$ can get more exact depth predictions compared with $t = 100$ or $t = \infty$. Because if we carefully compare Fig. 7(i) and Fig. 7(j), our depth enjoys less error on the surface (<2mm). On the other hand, the 'side effect' brought by our temperature setting is much less than the setting of $t = 1$. Thus, the idea of 'CLA in the early stages, REG in the latter stages' is reasonable and leads to better depth details and point cloud results as discussed in the main paper.

### C.4   Comparison of Different ViT Capacities

We further evaluate the performance of MVSFormer-P and MVSFormer with larger ViT backbones (base model) in Tab. 12. Interestingly, DINO-base achieves worse performance compared with DINO-small. We think that smaller DINO (Caron et al., 2021) used in MVSFormer-P enjoys better generalization, because the DINO backbone is fixed in MVSFormer-P due to the costly plain-ViT design. On the other hand, Twins-base can achieve full depth improvements compared with the small one, but improvements of point clouds are negligible. Obviously, point cloud metrics are more difficult to be improved. But we can still expect for good

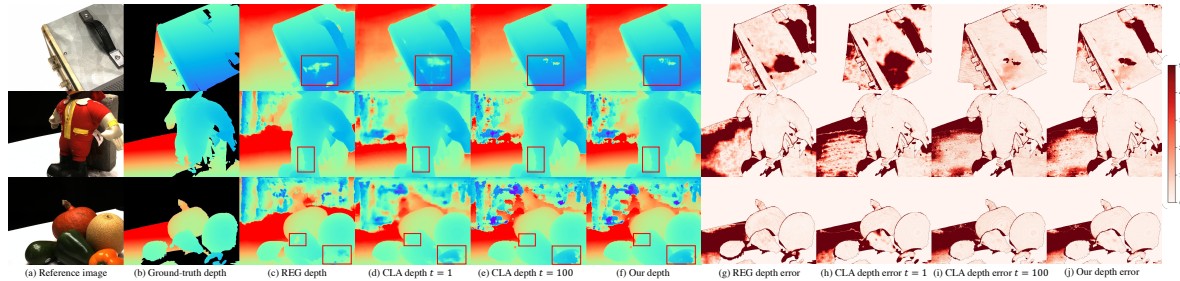

(a) Reference image  (b) Ground-truth depth  (c) REG depth  (d) CLA depth $t = 1$  (e) CLA depth $t = 100$  (f) Our depth  (g) REG depth error  (h) CLA depth error $t = 1$  (i) CLA depth error $t = 100$  (j) Our depth error

Figure 7: Qualitative depth comparisons among REG, and CLA with different $t$ settings. Our depth is based on $\{t_1, t_2, t_3, t_4 = 5, 2.5, 1.5, 1\}$. (g)-(j) indicate the depth error (mm).

performance from larger trainable pre-trained ViTs. Besides, both DINO-base and Twins-base make training be converged faster compared with small ones.

Table 12: Ablations about different capacities of MVSFormer-P and MVSFormer.

|  | $e_2 \downarrow$ | $e_4 \downarrow$ | $e_8 \downarrow$ | Acc.$\downarrow$ | Cop.$\downarrow$ | Ovl.$\downarrow$ |
|---|---|---|---|---|---|---|
| DINO-small | 17.18 | 11.96 | **8.53** | 0.327 | 0.265 | 0.296 |
| DINO-base | 17.41 | 12.22 | 8.67 | 0.334 | 0.268 | 0.301 |
| Twins-small | 17.50 | 12.48 | 9.14 | 0.327 | **0.251** | **0.289** |
| Twins-base | **16.78** | **11.94** | 8.82 | **0.326** | 0.252 | **0.289** |

## C.5  The Performance of Pre-trained ResNet50 with All Other Techniques

To further ensure the effectiveness of pre-trained ViTs for MVS, we provide more results in Tab. 13 about pre-trained ResNet50 with all other techniques used in our MVSFormer including the multi-scale training. Since CNNs enjoy good spatial invariance, the multi-scale training for ResNet50 is not as important as one for ViT based MVSFormers. The external experiment in Tab. 13 shows that multi-scale training can not improve ResNet50 a lot.

Table 13: Results of ResNet50 with all other proposed components. 'T-CLA' indicates temperature based depth with CLA.

| Pre-trained | Multi-scale | T-CLA | $e_2 \downarrow$ | $e_4 \downarrow$ | $e_8 \downarrow$ | Overall$\downarrow$ |
|---|---|---|---|---|---|---|
| ResNet50 |  |  | 20.09 | 15.11 | 11.78 | 0.323 |
| ResNet50 | ✓ | ✓ | 20.38 | 13.87 | 10.37 | 0.312 |
| DINO-small | ✓ | ✓ | **17.18** | **11.96** | **8.53** | 0.296 |
| Twins-small | ✓ | ✓ | 17.50 | 12.48 | 9.14 | **0.289** |

## C.6  More Detailed Quantitative Analysis of REG, CLA, and Temperature-based CLA

**Quantitative analysis.** Here we first re-list these important results in Tab. 14 from Tab. 6 for a clear and concrete illustration. The Accuracy (Acc) of point clouds indicates the mean value of reconstructed points whose closest distance to the ground-truth points. So 'Acc' can be seen as the precision of reconstructed points from the model. On the other hand, 'Cop' shows how complete the proposed method reconstructs. From Tab. 14, compared with REG, our temperature based depth from CLA can achieve better Acc and Ovl, with similar Cop, which means that our CLA enjoys better confidence maps that can filter outliers near the boundaries successfully with more precise point clouds. Note that vanilla CLA fails to get good Acc (0.349 vs 0.336) with similar Cop (0.248 vs 0.249). It means that the points reconstructed by inexact argmax from vanilla CLA are not precise (vanilla CLA does not produce more points because Cop are similar), which can be greatly solved by our temperature based depth.

Table 14: Quantitative point cloud results of REG, CLA, and our temperature-based CLA of MVSFormer.

|  | Accuracy (Acc) mm | Completeness (Cop) mm | Overall (Ovl) mm |
|---|---|---|---|
| REG | 0.336 | 0.249 | 0.293 |
| vanilla CLA | 0.349 | **0.248** | 0.298 |
| ours | **0.327(-0.009)** | 0.251(+0.002) | **0.289(-0.004)** |

**More ablations about temperatures.** We provide more ablations for the temperature setting in Tab. 15 for a deeper discussion about REG and CLA. In Tab. 15, the 1mm depth error is further compared, which shows that decreasing the temperature for the last stage can effectively get the exact depth with lower 1,2mm errors, while the training and inference disparity of the last stage is negligible. Moreover, the setting of $\{\infty, \infty, \infty, 1\}$ achieves the best depth metric and overall point cloud reconstruction, but the gap of accuracy and completeness is larger than $\{5, 2.5, 1.5, 1\}$. Thus slightly regressing in the early stages is still useful for the reconstruction with higher precision. Note that $\{5, 5, 5, 1\}$ outperforms $\{5, 2.5, 1.5, 1\}$ in most metrics except the completeness, but the improvement is not obvious. This phenomenon indicates that our critical idea of making early stages work as CLA and latter stages work as REG is validated; and adjusting the depth prediction for the last stage enjoys much more benefits and less cost compared with other stages.

Table 15: Quantitative comparisons for different temperature settings of MVSFormer.

| $t$ | $e_1 \downarrow$ | $e_2 \downarrow$ | $e_4 \downarrow$ | $e_8 \downarrow$ | Accuracy$\downarrow$ | Completeness$\downarrow$ | Overall$\downarrow$ |
|---|---|---|---|---|---|---|---|
| $\{\infty,\infty,\infty,\infty\}$ | 32.65 | 18.03 | 12.32 | 8.94 | 0.3488 | 0.2480 | 0.2984 |
| $\{1, 5, 5, 5\}$ | 31.75 | 18.73 | 13.34 | 9.83 | 0.3401 | 0.2511 | 0.2956 |
| $\{5, 1, 5, 5\}$ | 31.49 | 18.07 | 12.53 | 9.08 | 0.3475 | 0.2483 | 0.2979 |
| $\{5, 5, 1, 5\}$ | 32.39 | 17.99 | 12.37 | 8.98 | 0.3529 | 0.2546 | 0.3069 |
| $\{5, 5, 5, 2\}$ | 28.62 | 17.32 | 12.22 | 8.93 | 0.3465 | **0.2364** | 0.2915 |
| $\{5, 5, 5, 1.5\}$ | 27.87 | 17.26 | 12.22 | 8.94 | 0.3427 | 0.2376 | 0.2902 |
| $\{5, 5, 5, 1\}$ | 26.99 | 17.23 | 12.24 | 8.95 | **0.3265** | 0.2513 | 0.2889 |
| $\{\infty, \infty, \infty, 1\}$ | **26.67** | **17.18** | **12.19** | **8.91** | 0.3381 | 0.2390 | **0.2886** |
| $\{5, 2.5, 1.5, 1\}$ | 27.28 | 17.50 | 12.48 | 9.14 | 0.3270 | 0.2512 | 0.2891 |

## C.7 Ablations about More Source Views

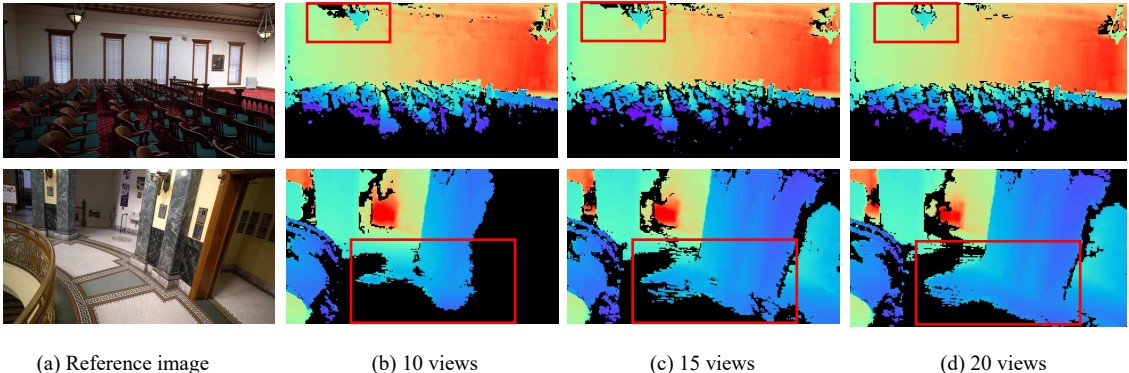

(a) Reference image     (b) 10 views     (c) 15 views     (d) 20 views

Figure 8: The view number ablation. All visualized depths are filtered by the confidence map $> 0.4$.

During the testing on Tanks-and-Temples (Knapitsch et al., 2017), we find that camera poses from the advanced set are very challenging. Therefore we try to expand source views with candidates of the selected source views which have not been included according to the view selection from Yao et al. (2018). We find that increasing the number of source views can effectively improve the performance in complex scenes of Tanks-and-Temples. Thanks to the visibility normalization (Zhang et al., 2020; Giang et al., 2021) used in MVSFormer, our method can be generalized to $N = 15, 20$, and achieve better results on Tanks-and-Temples.

As shown in Fig. 8, depth maps predicted by 20-view inputs are more reliable than ones from 10-view. Quantitative results from Tab. 16 show that the state-of-the-art results on Tanks-and-Temples from our main paper can be further improved with more source views.

Table 16: Ablation studies on the number of total views (reference and source views).

| | Intermediate | | | | | | | | | Advanced | | | | | | |
|---|---|---|---|---|---|---|---|---|---|---|---|---|---|---|---|---|
| | Mean | Fam. | Fra. | Hor. | Lig. | M60 | Pan. | Pla. | Tra. | Mean | Aud. | Bal. | Cou. | Mus. | Pal. | Tem. |
| 10-view | 64.90 | 81.85 | 66.29 | 59.47 | 65.03 | 64.69 | 62.18 | **61.44** | 58.22 | 39.55 | **28.28** | 45.60 | 36.90 | 50.05 | 33.99 | 42.46 |
| 15-view | 65.89 | 81.32 | 68.54 | **60.59** | 67.82 | 64.59 | 63.70 | 61.19 | 59.34 | 40.19 | 28.19 | 45.59 | 38.64 | 51.91 | 33.79 | **43.01** |
| 20-view | **66.37** | **82.06** | **69.34** | 60.49 | **68.61** | **65.67** | **64.08** | 61.23 | **59.53** | **40.87** | 28.22 | **46.75** | **39.30** | **52.88** | **35.16** | 42.95 |

### C.8 Qualitative Results of DTU

We provide qualitative results of DTU compared with CDS-MVSNet (Giang et al., 2021) and GBiNet (Mi et al., 2021). Qualitative depth and confidence comparisons are shown in Fig. 9, while point clouds are compared in Fig. 10. From Fig. 9, our MVSFormer can achieve more robust depth maps compared with others. Notably, depth maps from MVSFormer are as smooth as the regressive depth got from CDS-MVSNet. Besides, the *argmax* operation used in GBiNet fails to achieve stable depth predictions, and relies heavily on confidence maps to filter invalid depth. But our MVSFormer can get not only good depth predictions but also reliable confidence maps, which is benefited from the proposed temperature-based depth prediction. From Fig. 10, our method can faithfully reconstruct some challenging point clouds, which are usually omitted by other competitors.

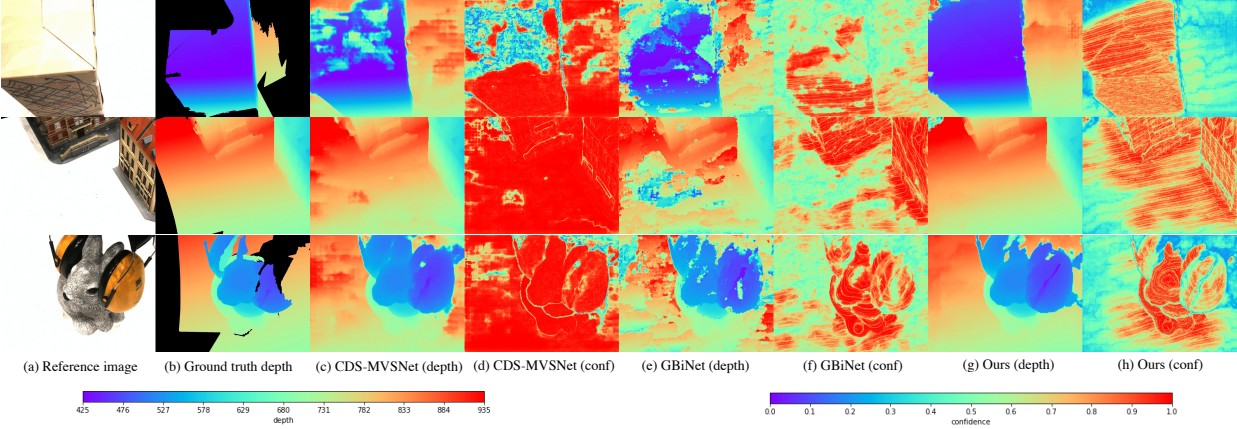

Figure 9: Qualitative DTU depth and confidence compared with CDS-MVSNet (Giang et al., 2021), GBi-Net (Mi et al., 2021), and our MVSFormer.

### C.9 Qualitative Results of Tanks-and-Temples

Our qualitative depth results of Tanks-and-Temples are shown in Fig. 11. Benefited by the proposed temperature-based depth prediction, MVSFormer can achieve not only good depth predictions but also reliable confidence maps, which leads to high-quality filtered depth in Fig. 11(d).

We also provide qualitative results of Tanks-and-Temples compared with TransMVSNet (Ding et al., 2021) and UniMVSNet (Peng et al., 2022). Fig. 12 shows qualitative results of 'Horse' and 'Lighthouse' in the Tanks-and-Temples intermediate set. From Fig. 12, our MVSFormer can reconstruct more details (better Recall) and generate point clouds with more accurate positions (better Precision). But TransMVSnet misses some structures in 'Horse' and predicts biased points in 'Lighthouse'. On the other hands, UniMVSNet suffers from many outliers in 'Horse', and fails to reconstruct a correct lighthouse.

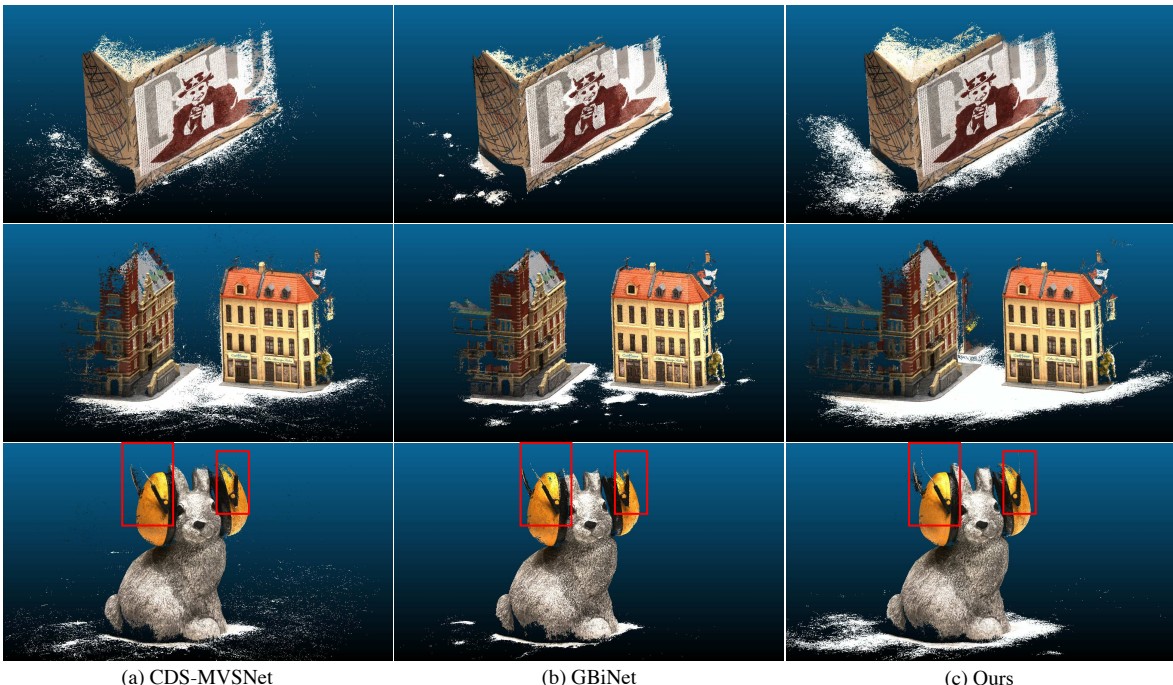

|  |  |  |
|---|---|---|
| (a) CDS-MVSNet | (b) GBiNet | (c) Ours |

Figure 10: Qualitative DTU point clouds compared with CDS-MVSNet (Giang et al., 2021), GBiNet (Mi et al., 2021), and our MVSFormer. Please zoom-in for details.

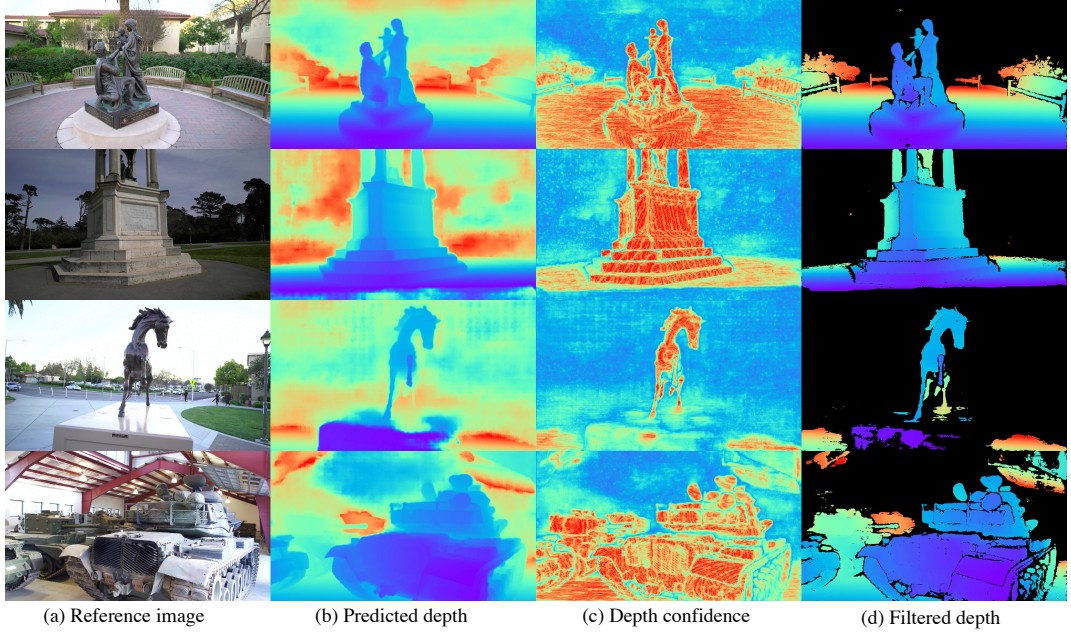

|  |  |  |  |
|---|---|---|---|
| (a) Reference image | (b) Predicted depth | (c) Depth confidence | (d) Filtered depth |

Figure 11: Depth prediction, depth confidence, and filtered depth of our MVSFormer on Tanks-and-Temples. Depth maps are filtered by confidence > 0.5 in (d).

### C.10    Training Randomness of MVSFormer

To evaluate the training randomness of MVSFormer, we additionally train the other 4 Twins based MVSFormers and DINO based MVSFormer-Ps as in Tab. 17 with the same settings except for random seeds. The reconstruction results of MVSFormer and MVSFormer-P are stable.

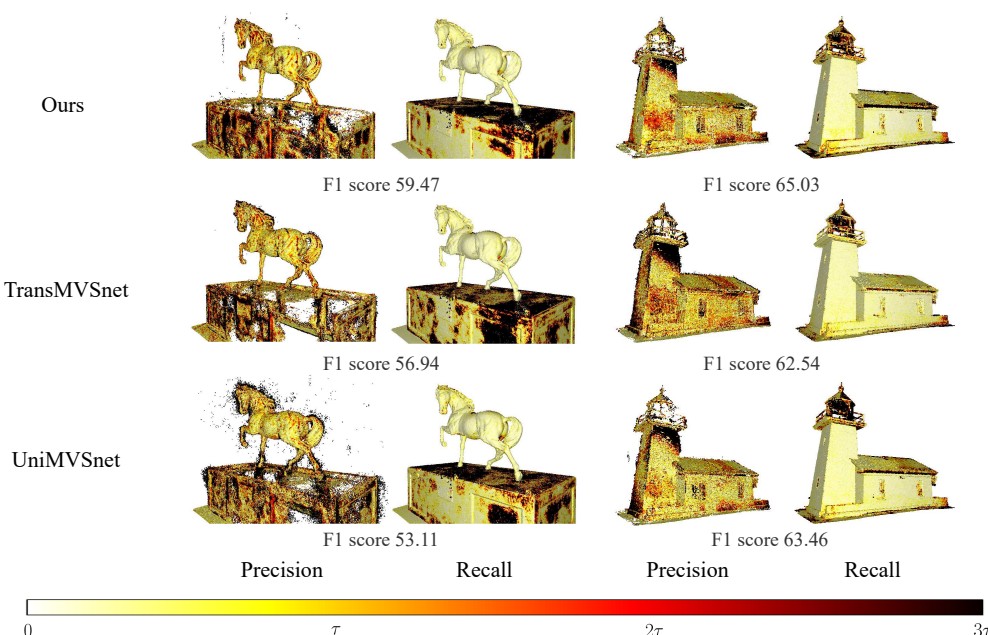

Figure 12: Qualitative results of Tanks-and-Temples (Horse and Lighthouse), compared with TransMVS-Net (Ding et al., 2021), UniMVSNet (Peng et al., 2022), and our MVSFormer. $\tau$ is the distance threshold provided officially. $\tau = 3$mm and $\tau = 5$mm for 'Horse' and 'Lighthouse' respectively.

Table 17: Training randomnesses of MVSFormer and MVSFormer-P. The std deviations are after the $\pm$.

| Methods | Accuracy(mm)↓ | Completeness(mm)↓ | Overall(mm)↓ |
|---|---|---|---|
| MVSFormer | 0.3274±6.36e-4 | 0.2518±5.31e-4 | 0.2896±3.73e-4 |
| MVSFormer-P | 0.3275±7.93e-4 | 0.2652±8.26e-4 | 0.2964±3.15e-4 |

## D   Real-World Results

We show some challenging real-world cases in Fig. 13 compared with CasMVSNet (Gu et al., 2020). Colmap (Schönberger et al., 2016) is utilized for camera poses. We use the same setting of Gipuma fusion (Galliani et al., 2015) with disparity threshold 0.2, number consistent 3 for both CasMVSNet and MVSFormer. The confidence thresholds are set as 0.9 and 0.5 for CasMVSNet and MVSFormer respectively, which follow their DTU settings. From Fig. 13, CasMVSNet suffers from messy outliers without segmentation masks, while our MVSFormer can achieve good results. Besides, our method can get robust results for texture-less and reflective objects.

## E   More Point Cloud Results

We show all point clouds of DTU test set generated by our MVSFormer in Fig. 15. And point clouds of both intermediate and advanced sets of Tanks-and-Temples are shown in Fig. 16. Moreover, ETH3D point clouds are shown in Fig. 14.

## F   Limitations and Future Works

We discuss the limitations and potential future works. In particular, 1) we utilize the recent ViTs pre-trained with self-supervised (MAE (He et al., 2021), DINO (Caron et al., 2021)) and supervised (Twins (Chu et al., 2021a)) tasks, while it is an interesting future work of exploring the influence of different pre-training tasks on MVS. The involved methods – MAE, DINO, and Twins are relatively representative. Specifically, MAE and DINO are based on the vanilla plain-ViT, while Twins is based on the hierarchical-ViT. Additionally,

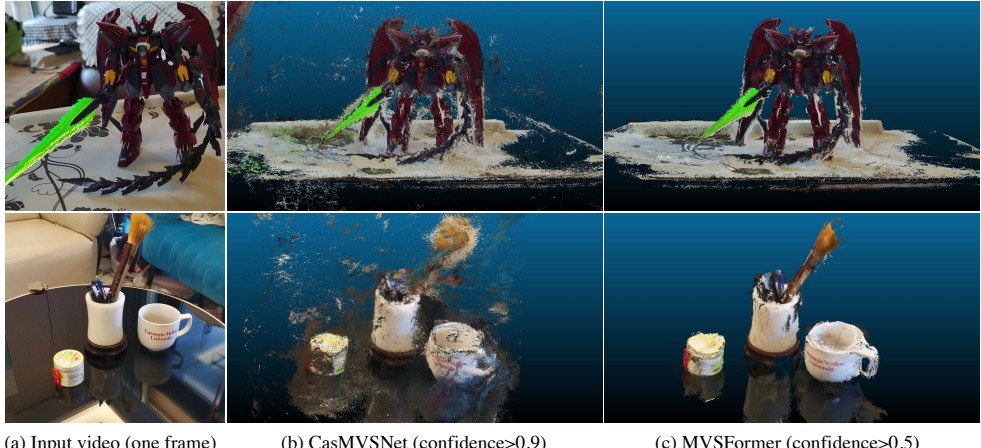

|  (a) Input video (one frame)  |  (b) CasMVSNet (confidence>0.9)  |  (c) MVSFormer (confidence>0.5)  |

Figure 13: Point Clouds of two real-world cases without masking compared with CasMVSNet (Gu et al., 2020). These cases are challenging in complex structures (the first row), hairy, texture-less and reflective objects (the second row).

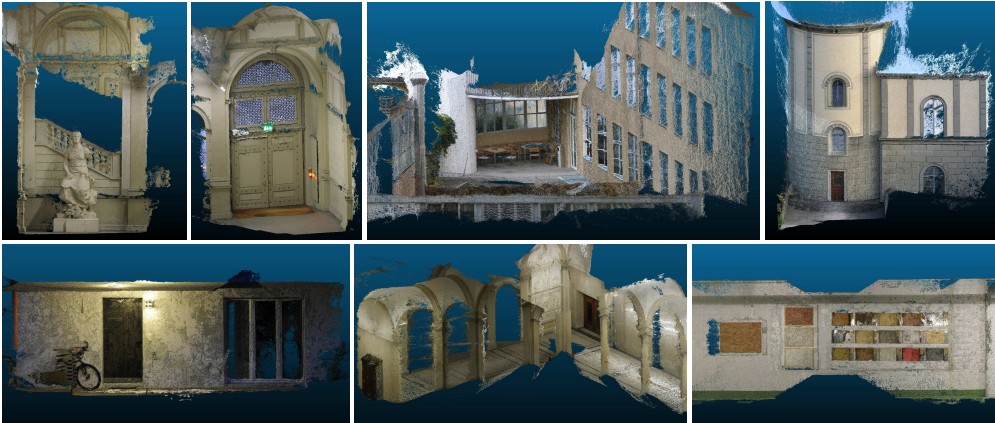

Figure 14: Point Clouds of scene dataset ETH3D (Schops et al., 2017) reconstructed by MVSFormer.

note that due to the memory cost limitation, we have to freeze ViT weights in MVSFormer-P, which loses some advantages over another variant – Twins based MVSFormer. Therefore, these ViTs are not ensured to be pre-trained with the same architecture and training settings in our paper. So it is interesting to use the same ViT architecture for different pre-training tasks, and further explore how these pre-training tasks (supervised and self-supervised) affect MVS. However, re-training all of these ViTs with different pre-training tasks is very non-trivial, and demands extraordinary computing resources. Thus we take it as future work. Critically, the performance of our proposed models has already outperformed all existing methods; this demonstrates the efficacy of our models. 2) Despite being simple, the fusion method used in this paper is good enough to make our models competitive. Here, we only consider the single and multi-scale feature addition. On the other hand, more technical and informative fusion strategies would be much advisable, such as cross-attention (Vaswani et al., 2017) and GRU modules (Cho et al., 2014). This however would also be a very interesting future work that may inspire the community.

## G    Broader Impact

Our methods can take the 3D reconstruction based on 2D images. Since learning-based MVS methods can be generalized to various real-world datasets, the proposed method may cause some societal impacts with controversial 2D images. Note that we only provide technical methods in this paper, but the real-world practice with potential negative societal impacts should be further considered.

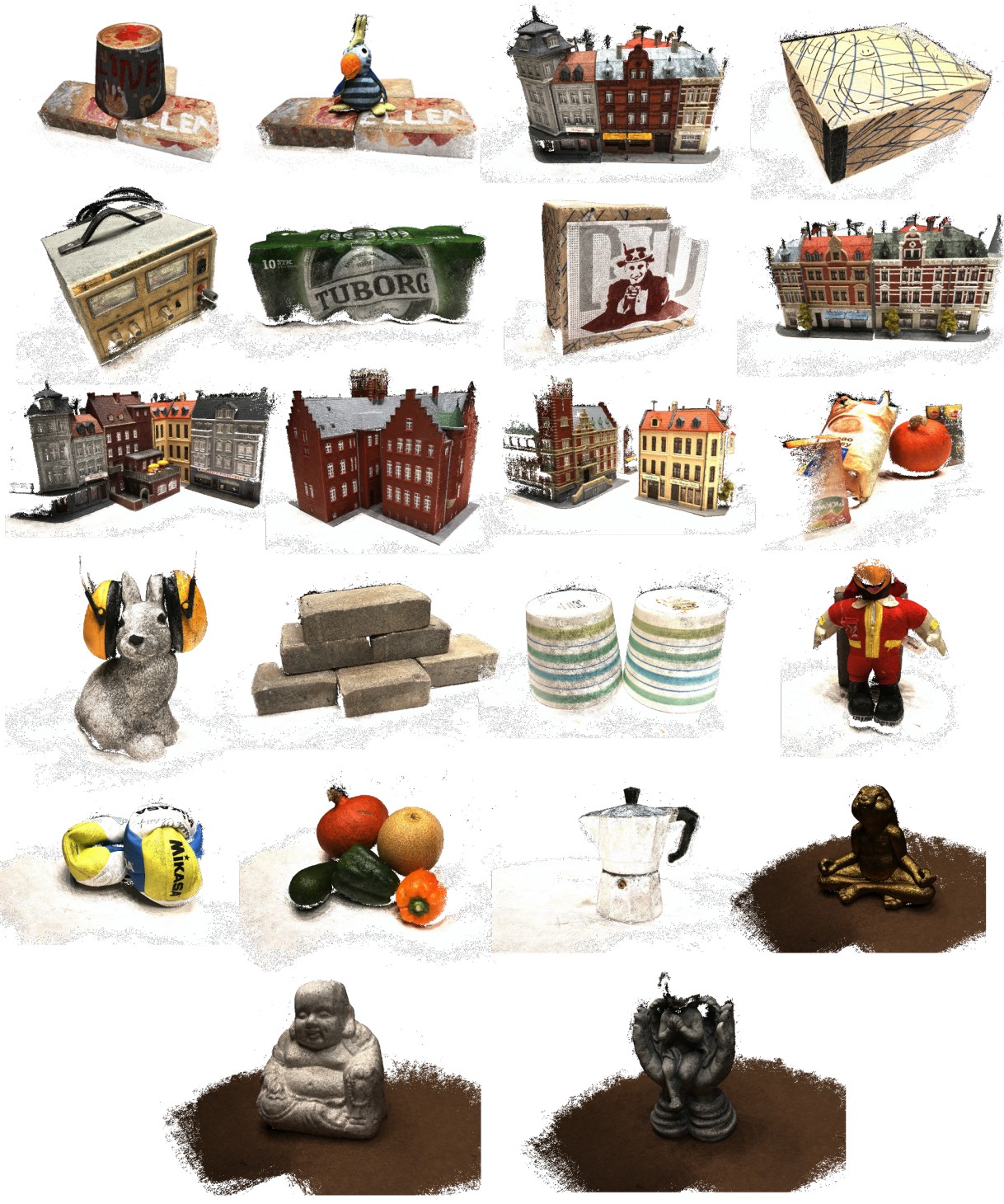

Figure 15: Point Clouds of all test set in DTU (Aanæs et al., 2016) reconstructed by MVSFormer.

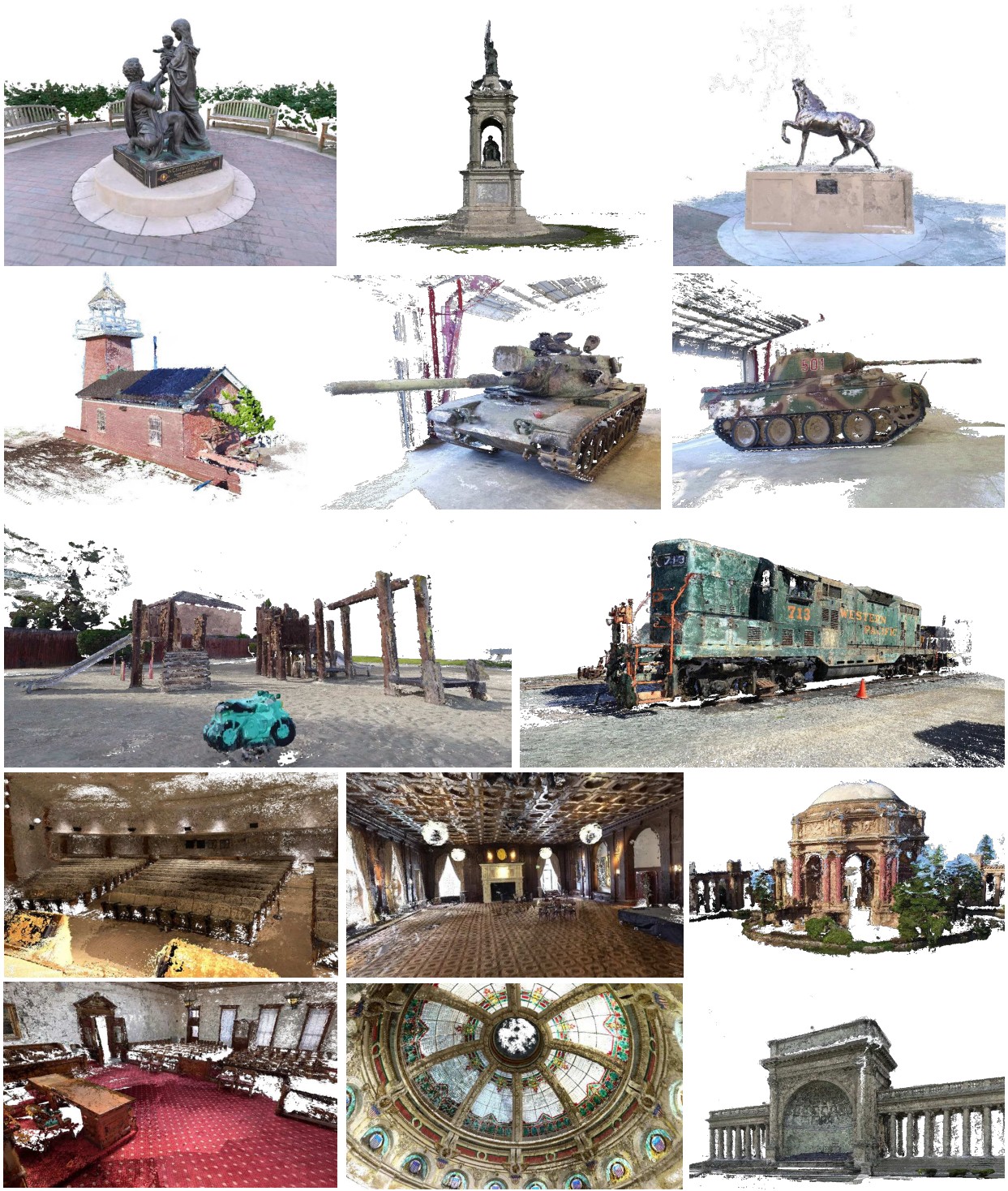

Figure 16: Point Clouds of Tanks-and-Temples (Knapitsch et al., 2017) reconstructed by MVSFormer.

