# OpenReview forum: "MVSFormer: Multi-View Stereo by Learning Robust Image Features and Temperature-based Depth"
_TMLR — Accepted by TMLR_

### Review · Reviewer_HVmc · 2022-10-04

**Summary Of Contributions:**

This paper introduces a new deep learning model for multi-view stereo (MVS) achieving state of the art results on Tank & Temples and DTU, the two reference benchmarks for this family of models. A MVS model can be subdivided into three steps: (1.) feature extraction from multiple views, (2.) feature matching to create a cost volume and (3.) feature refinement. The authors of this work take off the shelf solutions for step 2 and 3 while proposing a new model to improve step 1. Key of their contribution is the use of an hybrid convolutional + transformer architecture that can incorporate both local and high resolution details (through the convolutional path) as well as global context (through the transformer path) into extracted features. To make the training more efficient the authors propose to reuse pre-trained ViT for other self-supervised learning tasks. To improve the model predictions the authors also introduce a temperature scaling schema to sharpen or smoother the model output at inference time, further improving performance. The sum of all these contributions allows this work to reach state of the art results.

**Broader Impact Concerns:**

The paper contains a short broader impact section, I don't feel like the authors would need to comment more on this topic as the proposed method is rather technical and has no evident direct broader impact on society.

**Requested Changes:**


*Critical for acceptance*: Please try to improve some issues highlighted in weaknesses 1 and expand the explanation of how the confidence in the prediction is computed (weaknesses 2).

*Optional to strengthen submission*: quantify differences in uncertainty estimation between REG and CLA, motivate better the choice of the 4 temperature discussed in weaknesses 3, add std dev as discussed in weakness 4


**Strengths And Weaknesses:**

## Strengths

+ The ablation study of the paper is very well done and answers all the questions that I had come up with reading the method section of the paper. The authors carefully compare all the possible configurations of their rather complex system and motivate experimentally the choices made.

+ I found the insights on the different behavior of the confidence estimation for regression and classification trained disparity estimation systems quite interesting and providing insights that can be useful for other researchers not strictly working on MVS.

+  Performance of the model is solid and clearly outperforms previous solutions. Moreover the proposed additions don’t dramatically slow down the baseline as reported in Tab. 8.

## Weaknesses

1. Presentation of the paper could be improved on some aspects:
    - There are some grammatical mistakes throughout the document
    - Citations style does not match the way the text is written, for example in the related work section there are references to works that reads something like “Xu et al Xu et al. (2020)”. Please adapt the manuscript to this citation style.
    - I found the presentation of both MVSFormer-P and MVSFormer-H to be slightly confusing and adding an unnecessary amount of information. I would suggest to keep *MVSFormer-H* as the main proposal of the work and use P as an ablation study on alternative transformer architectures to incorporate. H seems to outperform P in all the benchmarks considered.
    - I would add to all the ablation study a row/column in the tables corresponding to the full model, hust to compare the performance of the different ablations to the full model proposed.

2. I found the insights of this work on the different behavior of the CLA and REG losses in terms of predicting confidence to be quite interesting, but I would like the authors to have digged a little deeper in this topic. In particular the evaluation right now it’s purely empirical and qualitative, a more quantitative one would have been preferable. Also from the text it’s not exactly clear how the confidence is calculated to obtain the visualization in Fig. 4. I would suggest to the author to include these details to make the manuscript more self-contained. It’s probably outside the scope of this manuscript but the analysis the authors have started is basically connected to how well the model can explain the uncertainty in its prediction. The uncertainty estimation literature is rich of methods developed for this tasks and if this is a strong requirement of the work the authors might also consider to discuss among future development the incorporation of a network formulation designed to estimate uncertainty (like the one proposed in [a])

3. Using temperature scaling only at inference time and not at training time seems slightly hacky. In practice the predictions of the model are oversharpened or smoothed based on some  empirical rules but an extensive study is not proposed on why this should be beneficial or needed.The proposed solution also creates discrepancy between train and testing regime which is usually something that should be avoided.
Also the 4 values picked for the temperature scaling at the 4 resolutions are quite arbitrary and not super well motivated. Tab. 5 shows a comparison to picking a single value for `t` at all scales but does not compare against picking different combinations of `t` at different scales.

4. The performance of the method and competitors/ablations is quite close. Adding an estimation fo the std deviation on the final performance introduced by the randomness of the training process would have made the submission stronger.

## References

a. Kendall, Alex, and Yarin Gal. "What uncertainties do we need in bayesian deep learning for computer vision?." Advances in neural information processing systems 30 (2017).

---

> ### Author Response · Authors · 2022-10-17
> **Response to Reviewer HVmc Part1**
>
> We appreciate the reviewer HVmc for all patient and valuable comments. We also revise related parts in the main paper in red.
>
> **1. The presentation could be improved: (1) grammatical mistakes; (2) citation problems; (3) keeping MVSFormer-H as the main method, and using -P as an ablation study; (4) adding to all the ablation studies a row/column for the full model.**
>
> Thanks for the constructive advice. For (1) and (2), we have carefully improved the grammar and citation style in our paper.
>
> For (3), sure, we agree with the reviewer’s suggestion that the presentation of both MVSFormer-P and MVSFormer-H is a little confusing. But MVSFormer-H does not outperform -P in all benchmarks. As in Tab3, -P enjoys better depth compared with -H. Moreover, -P is more efficient to be trained with a larger batch size (Tab6) and has fewer trainable parameters (Tab8). To make the presentation clearer, we keep the Twins-based MVSFormer as our default method without additional explanations in the revised paper (page5 Sec 3.1). And the DINO-based MVSFormer-P now is the alternative version with competitive performance and better efficiency (page6 Sec 3.1).
>
> For (4), we have already reported the full model performance in Tab3. Moreover, we highlight the full model results in Tab4 (green: MVSFormer-P, blue:MVSFormer-H).
>
> **2. (1) More quantitative evaluation for CLA and REG. (2) How is confidence calculated? (3) Discussion about the uncertainty estimation.**
>
> Thanks for the good comment. For (1), we have already provided qualitative analysis in Tab5 and Tab13 (C.6 of the Appendix). In particular, the concrete analysis of CLA and REG is discussed in C.6 of the Appendix. Compared with REG, CLA produces better confidence but suffers from inferior accuracy during the reconstruction, which is caused by inexact depth prediction. Furthermore, our temperature based depth from CLA can achieve better accuracy and overall, with similar completeness, which means that our CLA enjoys better confidence maps that can filter outliers near the boundaries successfully with more precise point clouds.
>
> For (2), we additionally provide the way to get confidence maps (CLA and REG) in A.4 of the Appendix. We get the CLA confidence with the highest confidence value of the predicted depth for each stage, while REG’s confidence is obtained after the local avg pooling as MVSNet. Then the final confidence map is averaged of ones from all stages.
>
> For (3), we thank the reviewer for this constructive insight. We cited the mentioned paper and added some discussion (the end of A.5 in the Appendix).  In terms of the definitions from the recommended citation, we think that both CLA and REG should be categorized as aleatoric uncertainty rather than epistemic uncertainty. Because we have not tested these methods with model uncertainty (e.g., dropout inference). More importantly, both CLA and REG only show the uncertainty of noise in the training distribution, rather than the Out-of-data examples. For example, most learning based MVS methods (both CLA and REG) demand a certain depth range before the prediction; and they cannot provide proper results with wrong depth ranges. However, given a certain depth range, CLA-based cascade MVS enjoys much more reasonable confidence as discussed in Sec 3.4, which is important for the reconstruction (Fig6). Finally, we still think that uncertainty estimation is an interesting problem in MVS as the future work.

---

> ### Author Response · Authors · 2022-10-17
> **Response to Reviewer HVmc Part2**
>
> **3. (1) The temperature is hacky. Extensive studies about the temperature depth prediction. (2) Does this method have a discrepancy between train and testing? (3) Picking different combinations of t and the motivation of different t.**
>
> Thanks for this comment. (1) Our paper actually provides an intuitive understanding and motivation about temperature. The temperature is not hacky, nor data-dependent. Specifically, all datasets are tested in the same setting (5,2.5,1.5,1) as mentioned in the main paper. Except for Sec 4.2, we have also provided extensive studies about the temperature in C.6 of the Appendix. The temperature is needed to achieve exact depth prediction for CLA with more precise point clouds (Tab13). Moreover, we provide a more detailed qualitative analysis about the temperature in C.3 of the Appendix to illustrate why temperature is useful. The temperature based depth is more precise and enjoys less error on the surface (<2mm) (C.3 of the Appendix).
>
> (2) Thanks for this question. We add a discussion to the end of Sec 3.4 about this point, which means that the core idea of “classify first, then regress” has negligible training and testing discrepancy. Although adjusting the temperature during the testing may suffer from some implications of the discrepancy between the train and test stages, we only regress the latter stages with only a few nearby depth hypotheses. Thus such a gap is largely narrowed. Besides, we also show that our training and testing discrepancy is small in Fig7.
>
> (3) Thanks for this question. The motivation to use different t is well indicated in Sec 3.4, i.e., “For early stages with low-resolution, we set larger t to make the model work as a CLA for a better global distinguishing ability. And for later stages with high-resolution, our model tends to use lower t as a REG to smooth local details”.
> To evaluate the effect of different combinations of t, we add more ablations in Tab14; and provide more related analysis in the Appendix. We further compare the 1mm depth error in Tab14 and find that regressing the last stage (t=1) in CLA can remarkably improve the exact depth prediction. Moreover, the setting of {$\infty,\infty,\infty$,1} achieves the best depth metric and overall point cloud reconstruction, but the gap of accuracy and completeness is larger than our {5,2.5,1.5,1}. Thus slightly regressing in the early stages is still useful for precise reconstruction. Although {5,5,5,1} slightly outperforms {5,2.5,1.5,1} except the completeness, the improvement is not obvious. Our critical idea of making the early stages work as CLA and the latter stages work as REG is still validated. These discussions are also considered in the C.6 of the Appendix.
>
> **4. Adding std deviation for the final model.**
>
> Thanks for this comment. We now provide the std deviation for our full models (MVSFormer-H and MVSFormer-P) with additional 4 results trained with different random seeds in C.10 of the Appendix. The reconstruction results of MVSFormers are stable.

---

### Review · Reviewer_QxQb · 2022-10-07

**Summary Of Contributions:**

Authors develop a framework for learning-based multi-view stereo, that incorporates image features calculated by a vision transformer (ViT). Based on existing approaches that construct a cost volume from FPN features, they augment the FPN with additional feature channels derived from two pretrained ViT models. There is a detailed evaluation showing which ViT variant performs best. State-of-the-art MVS performance is achieved on Tanks & Temples. There are also some smaller innovations – use of a loss that can interpolate between existing regression-based and classification-based training of the depth predictions, and a novel multi-scale training strategy. Their benefits are evaluated in a detailed ablation study.

**Requested Changes:**

See above regarding ResNet50; I feel this is an important point that needs clarifying, as it is central to the message of the paper.

Typos / etc:
- p4: "strategy to ignite the ViTs for MVS" – ignite is maybe not the best word!
- p5: "bicuic" → bicubic
- p6: "with even frozen DINO of trainable GLU" – fix the grammar
- many citations are not correctly parenthesised (use \citep)
- appendix A title: change "implemented" to "implementation"
- p17: "NVIDA" → "NVIDIA"


**Strengths And Weaknesses:**

Overall this appears a solid paper, clearly written and well motivated, and exhibiting state-of-the-art results on MVS in certain settings.

Please clarify whether the ResNet50 baseline (tab. 3) was fine-tuned during MVS training (as with Twins), or kept frozen (as with DINO). If it was kept frozen, then fine-tuning it seems an important baseline to include (to definitively confirm it's the use of a ViT that benefits, not simply having more network capacity for feature extraction).

Aside from this, the evaluation seems comprehensive, and demonstrates the benefit of using a ViT as an additional input capturing 'context' in the feature extractor. Strong results are shown on two standard benchmarks – Tanks & Temples, and DTU. There is a detailed ablation study investigating the benefits of various aspects of the proposed system.

While the proposed framework is novel, its individual technical contributions are not very deep, and it is closely based on existing learnt-MVS pipelines. Use of ViT features seems a reasonable thing to try, and the paper makes a worthwhile contribution by investigating this in some detail – however there is no particular technical novelty. The proposed loss that interpolates between hard and expected depths is, again, reasonable (and investigating it is a worthwhile contribution), but not very technically substantial. The same is true of the multi-scale training scheme.

There is plenty of discussion of background and related work. The writing is fluent and readable throughout, with very few spelling/grammar issues.

---

> ### Author Response · Authors · 2022-10-17
> **Response to Reviewer QxQb**
>
> We appreciate the reviewer QxQb for all patient and valuable comments. We also revise related parts of the main paper in red.
>
> **1. Please clarify whether the ResNet50 baseline (Tab3) was fine-tuned during MVS training.**
>
> Thanks for this comment. We would clarify that our ResNet50 baseline is finetuned during the MVS training. We also add this information to Sec 4.2 paragraph1, i.e., “Note that both ResNet50 and Twins are trainable, while DINO and MAE are frozen”.
>
> **2. Typos, grammar problems, and many citations are not correctly parenthesized.**
>
> Thanks for these comments. We have carefully revised all these problems, especially for the citation style.
>
> **3. The contributions (ViT, temperature based depth, multi-scale) are novel and worthwhile, but not technically substantial.**
>
> Thanks for this comment. We should clarify that all our proposed contributions are practical and important to improve MVS learning in general. Extensive experiments also support the efficacy of our contributions; Furthermore, these techniques have not been pointed out in previous investigations.  And empirically they are useful to the community. In particular, there is no discussion about the influence of ViT and other pre-trained models in previous MVS works. We also discuss the merits and drawbacks of classification (CLA) and regression (REG) based MVS methods, and further propose a lightweight temperature-based depth prediction to unify their advantages.
> Last but not least, multi-scale training is also an important solution to release the potential of ViTs in MVS learning. We also find that the multi-scale training with only a few high-resolution images outperforms the costly high-resolution finetuning for MVSFormer as in Tab3. All codes of the proposed method will be released if accepted.

---

### Review · Reviewer_Yqpj · 2022-10-29

**Summary Of Contributions:**

The paper introduced a new multi-view stereo algorithm. The network formulation leverages the pre-trained ViT for feature extraction, and the proposed algorithm MVSFormer that leverages those learned features are able to significantly advance reconstruction accuracies on public benchmarks like DTU, etc. The paper contained several ablation studies on various part of the network formulations and experimental settings, and provided insights for designing accurate multi-view stereo algorithms.

**Broader Impact Concerns:**

This is a standard multi-view stereo paper and I see little ethical concerns.

**Requested Changes:**

I think the strength of this paper is strong, and I wish the code can be released as promised by the authors. However since this is a journal paper, it may be a good idea to at least verbally address some of the weakness I raised in the comments in the paper.

**Strengths And Weaknesses:**

[Strength]
- The formulation of the network architecture is relatively straight-forward and rather simple, but the performance gain is quite visible. The conclusions of this paper, especially claims in the feature extraction part, could be instrumental to the research community.
- Solid and strong experimental results on traditional and popular MVS benchmarks. The numbers and visual results are good.
- The overall presentation of the paper is good, with good descriptions of the approach as well as plenty of ablation studies for experiments.


[Weaknesses]
- The network relies on pre-trained ViT models and from the descriptions of the paper it appears that there are huge memory requirements of training this network. There is perhaps no solutions to actually address this weakness in the scope of this paper though, but I wonder if there are potential lighter versions of the network that could be easier to train. Or -- if this paper is actually getting better performance at the cost of a much complex architecture.
- The paper shows strong results on DTU and such. However, there are also a series of papers (DPSNet, normal-assisted stereo depth estimation, MVS2D, etc) who are also reporting results on scene datasets (as opposed to traditional object-centric datasets) like SUN3D, ScanNet, etc. It would be interesting to see if similar network designs also applies to those applications.
- In the Appendix B, table 8, the authors reports computational times of the approach, but didn't really compare with other concurrent MVS algorithms. It's a bit challenging to reference the relative gain/loss for speed.

---

> ### Author Response · Authors · 2022-11-02
> **Response to Reviewer Yqpj**
>
> **1. The concern about the memory requirements and the request for a lighter model.**
>
> Thanks for this concern.  We indeed consider the straightforward solution to the heavy computational cost. Particularly, one may want to check that the input image size for pre-trained ViTs is halved in this paper (Fig.2, Sec3.1) to save the computation and GPU memory. Since the ViT encoding tends to learn better high-level features (Sec.D.1), the halved inputs for ViT are still competitive for the MVS (Tab.3), while our FPN still takes full image sizes for good details.
> Furthermore, we have also provided MVSFormer-P, whose backbone DINO is frozen during the training as a lighter version of MVSFormer. MVSFormer-P also enjoys good performance compared with other SOTA MVS methods. Moreover, our proposed gradient accumulated multi-scale training is also efficient. The training memory costs are shown in Tab.6.
>
> **2. Getting better performance at the cost of a much more complex architecture?**
>
> On the other hand, we advocate "the good performance from the good feature representation (rather than purely complex architecture)".
> Our focus of this paper is to emphasize that except for the cost volume learning, a good MVS solution can also be achieved by architectural designs for feature extraction. This is slightly overlooked in previous works. Further, well-representative features can be easily pre-trained from the vanilla large-scale 2D image classification tasks. This largely reduces the costly MVS training (e.g., by freezing this pre-trained 2D ViT backbone).
>
> **3. Reporting results on scene dataset.**
>
> Thanks for this comment. In general, the setting of MVSFormer is following those in DTU and BlendedMVS, which is different from the one of ScanNet, e.g., view number. Essentially, our experiments in this paper are solid enough to validate our contributions and claims. On the other hand,  except for the Tanks-and-Temples, we would add more experiments in  Scene datasets. So we compare our method in ETH3D with a more challenging setting in Sec.C of the Appendix. ETH3D contains 13 training and 12 test scenes, which include both indoor and outdoor challenging scenes. We evaluate the ETH3D with the model trained on DTU and finetuned on BlendedMVS, which means that our method does not see any other scene data. As shown in Tab.9, our method still enjoys good robustness and outperforms other SOTA learning-based MVS methods. This further validates the efficacy of our model in scene scenarios. More qualitative results are shown in Fig.14 of the Appendix.
>
> **4. Comparing other concurrent MVS algorithms in the computation.**
>
> Thanks for this comment. To address it, we additionally compare three concurrent MVS methods in Tab.8 (CasMVSNet, CDS-MVSNet, TransMVSNet). Note that our baseline has more parameters as it contains 4-stage to fit the ViT architecture (the depth hypothesis number has been reduced to balance the computation). Although the three competitors enjoy less trainable parameters, some of them fail to achieve better efficiency compared with MVSFormer in GPU memory and speed.
> In particular, CasMVSNet costs the most memory without the group-wise pooling for the cost volume. The sophisticated TransMVSNet suffers from slow inference speed and large memory costs. CDS-MVSNet is slightly more efficient than MVSFormer, but it is discouraged in its performance. We have also added these discussions in Sec.B of the Appendix.

---

### Decision · Action_Editors · 2022-12-12

**Recommendation:** Accept with minor revision

**Comment:**

Given all three accept ratings, I am inclined to accept the paper.

The revisions have improved the paper and all three reviewers are in favor of acceptance. I am giving "minor revisions" with the aim of enabling the authors to address the final recommendation from Reviewer QxQb. I'm not positive these are visible to the authors (since I now see that the recommendations were not as they were coming in), so I am copying them here: ".. The extra experiments on ETH3D are appreciated (in fact I think it would be good if at least part of appendix C could be squeezed into the main paper), as is the clarification regarding how the ResNet-50 baseline is trained. ....".  Thus, the only change the authors might consider would be squeezing in Appendix C. This is, however, up to the authors. If they wish to keep them in the appendix, that is also acceptable to me.



**Audience:**

Yes, the paper covers the popular topic of multiview stereo using learning. This topic in general will be of clear interest to the community, and the reviewers commented that many of the particular insights from the paper will be of interest to others.

**Claims And Evidence:**

All three reviewers were in favor of accepting the paper, and stated that the paper was accompanied by clear experimental support.